# Bayesian model selection for complex dynamic systems

Christoph Mark[1], Claus Metzner[1], Lena Lautscham[1], Pamela L. Strissel[2], Reiner Strick[2] & Ben Fabry[1]

Time series generated by complex systems like financial markets and the earth's atmosphere often represent superstatistical random walks: on short time scales, the data follow a simple low-level model, but the model parameters are not constant and can fluctuate on longer time scales according to a high-level model. While the low-level model is often dictated by the type of the data, the high-level model, which describes how the parameters change, is unknown in most cases. Here we present a computationally efficient method to infer the time course of the parameter variations from time-series with short-range correlations. Importantly, this method evaluates the model evidence to objectively select between competing high-level models. We apply this method to detect anomalous price movements in financial markets, characterize cancer cell invasiveness, identify historical policies relevant for working safety in coal mines, and compare different climate change scenarios to forecast global warming.

---

[1] Department of Physics, Friedrich-Alexander University Erlangen-Nürnberg, Erlangen 91052, Germany. [2] Department of Gynecology and Obstetrics, University Hospital Erlangen, Erlangen 91054, Germany. Correspondence and requests for materials should be addressed to C.M. (email: christoph.mark@fau.de)

The movements of invasive cancer cells, price fluctuations of stocks, and the temperature fluctuations tied to global warming all represent random walks. In contrast to the well-known Brownian motion of pollen grains in water, the random walk of, e.g., stock prices cannot readily be described by a constant diffusion coefficient. Instead, the data follow a simple low-level model (e.g., a Gaussian distribution with a standard deviation that corresponds to the diffusion constant) on short time scales, but the model parameters can fluctuate on longer time scales according to a high-level model.

Statistical models in which the low-level parameters fluctuate randomly on long time scales or large spatial scales have been previously described by a superposition of statistical processes—commonly referred to as "superstatistics"[1,2]. For example, the non-Gaussian, fat-tailed distribution of stock returns can be described by a superposition of Gaussian distributions of varying standard deviation $\sigma$. A superstatistical analysis aims to reconstruct the distribution of $\sigma$ (how often the market is calm (small $\sigma$) or turbulent (large $\sigma$)) and its temporal auto-correlation (the "life-time" of different market conditions) from price fluctuations[3].

Superstatistics has been successfully applied to a variety of financial[4–6], environmental[7–9], social[10,11], and biological systems[12]. Moreover, current superstatistical methods can determine not only how frequently certain parameter values are realized but can also pin-point when parameter values change[13–16]. However, current methods lack the ability to objectively compare different time-varying parameter models. Without such an objective measure, one risks to either underestimate parameter fluctuations (and therefore to lose valuable information about the system's dynamics), overestimate parameter fluctuations (and therefore to mistake noise for signal) or assume the wrong type of parameter dynamics (e.g., assuming gradual parameter variations in the case of abrupt parameter jumps).

Two established approaches to infer time-varying parameters and their uncertainty are Monte Carlo methods[17–20] (which approximate parameter distributions by random sampling) and Variational Bayes techniques[21] (which approximate parameter distributions analytically by simpler distributions). However, both approaches cannot directly estimate the so-called model evidence—the probability that the measured data is actually generated by the model (although Variational Bayes methods can at least provide a lower boundary). An objective comparison between different models is therefore not possible.

We propose an alternative approach for the inference of time-varying parameter models. We exploit that many time series can be fitted by evaluating the contribution of each data point to the low-level parameter distribution in an iterative way, time step by time step. This allows us to breakdown a high-dimensional inference problem into a series of low-dimensional problems. Furthermore, if the number of time-varying parameters is relatively modest ($\lesssim 3$), parameter distributions can be represented on a discrete lattice, enabling us to efficiently compute the model evidence by adding up all probabilities.

In a previous study, we have applied a superstatistical method to characterize the heterogeneous random walk of migratory tumor cells, whereby the two low-level parameters of a random walk—cell speed and directional persistence—were allowed to vary in time according to an arbitrary user-defined magnitude[13]. Here, we extend this approach and automatically tune the high-level model, i.e., the magnitude of parameter variations, based on the model evidence. We also utilize the model evidence as a powerful tool in hypothesis testing by comparing different high-level models of varying complexity. We apply our method to problems of current interest from diverse areas of research in social science, finance, cell biophysics and climate research.

## Results

**Bayesian updating.** Bayesian statistics provides a mathematical framework for improving our estimates of low-level parameters (e.g., volatility) as new noisy data (e.g., price fluctuations) become available. Bayesian updating starts with a prior distribution that describes any knowledge about the low-level parameters before seeing the data. By multiplying the prior with the likelihood function evaluated at each data point, one obtains the (non-normalized) posterior distribution (Fig. 1a). The likelihood function is the low-level model and describes the conditional probability of the current data point, given the current parameter values and possibly also the past data points. The posterior distribution provides the most likely low-level parameter values (the mode of the distribution) and their uncertainty (the width of the distribution).

We use a grid-based implementation of the Bayesian updating method, which allows us to compute the normalization constant of the posterior distribution by summing over the parameter grid. This normalization constant is the so-called model evidence and directly represents the probability that the data has been generated by the model[22,23]. The model evidence provides a quantitative measure to compare the statistical power of different models, as it rewards good model fit to the data and penalizes the choice of too many free parameters. This can be viewed as an automatic implementation of "Occam's razor"[24].

**Inference of time-varying parameters.** For models with time-varying parameters, a new posterior distribution is needed for each time step (Fig. 1b). The possibility of a parameter change between two time steps can be implemented by a redistribution of the probabilities on the parameter grid. This redistribution is mathematically defined as a norm-conserving transformation. For example, to account for stochastic Gaussian parameter fluctuations between successive time steps, the posterior distribution is convolved with a Gaussian kernel. In this case, the variance of the Gaussian kernel represents a high-level parameter. To account for arbitrarily large, abrupt jumps of the time-varying parameters, the corresponding transformation assigns a minimal probability to all values on the parameter grid. Furthermore, deterministic low-level parameter changes over time such as a linear trend can be achieved by a time-dependent transformation that shifts the posterior distribution according to a predefined function.

If this transformation is applied only in the forward direction of time, for example in a prospective study in which new data points arrive in real-time, the resulting parameter estimates are based only on the information contained in past data points. However, in the case of a retrospective analysis, the transformation can be applied in both forward and backward direction of time such that the parameter estimates at each time step incorporate the information contained in all data points.

The model evidence of a time-varying parameter model is obtained by summing the probability values of the (unnormalized) posterior distribution over the parameter grid for each time step in the forward time direction, and multiplying the sums from all time steps. The obtained model evidence assesses the goodness-of-fit of both the low-level model and the transformation that is applied to the posterior distribution. The model evidence can thus be used to compare different (high-level) hypotheses about the (low-level) parameter dynamics. When we systematically compute the model evidence over an equally spaced grid of the high-level parameter values, we obtain the complete distribution of the high-level parameters. By summing all model evidence values over the high-level parameter grid, we obtain a compound model evidence value that takes into account the uncertainty of the high-level parameters.

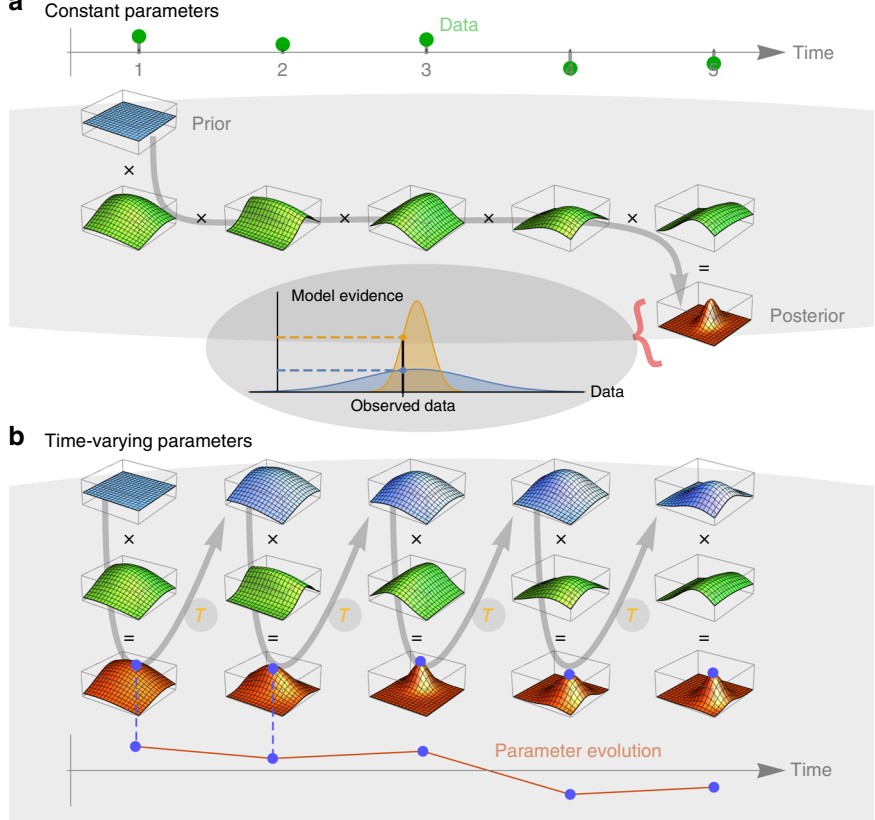

**Fig. 1** Bayesian updating and Occam's razor. **a** For a model with constant parameters, the multiplication of the prior distribution (blue) with all factors of the likelihood function (green; one factor for each new data point) directly yields the unnormalized posterior distribution (red). The normalization constant of the posterior distribution represents the model evidence. Here, restrictive models (inset, yellow) concentrate their evidence on a smaller subset of possible datasets compared to more flexible models (inset, blue) and thus attain a higher model evidence, if both models fit the data equally well. **b** For a model with time-dependent parameters, the multiplication of the prior distribution (blue) and the likelihood function (green) is carried out for each time step individually. Between time steps, the posterior distribution (red) is transformed according to a high-level model T, to reflect possible changes of the low-level parameters

The grid-based evaluation of the model evidence is a major advantage of this method, but it also limits the number of time-varying parameters to $\lesssim 3$ and the number of data points in the time series to $\lesssim 10^4$ as computation time and computer memory space increases linearly with the number of grid points times the number of data points. Within this niche, however, our method outperforms the state-of-the-art Hamiltonian Monte Carlo approach for inferring stochastic parameter variations (Supplementary Fig. 1).

**Policy assessment in coal-mining safety**. The number of coal-mining disasters in the United Kingdom between 1852 and 1961 in which ten or more men were killed has served as a classic example of a so-called change-point analysis[25–30] (Supplementary Data 1). At some point between 1880 and 1900 (the change-point), the number of disasters dropped abruptly to one third, due to updated safety regulations. Here, we use our method to identify which safety regulations have led to the significant decrease of mining disasters.

The statistics of the annual disaster count is readily described by a Poisson distribution with a single parameter—the accident rate. To identify the time point at which the accident rate changed systematically, we compare two competing models: The "classical" approach used in most studies assumes that the accident rate is piecewise constant before and after the change-point (Fig. 2a). Here, the Poisson process represents the low-level model, and the time point at which the accident rate changes is the high-level

parameter. Alternatively, we further allow for smaller Gaussian fluctuations of the accident rate over time (Fig. 2b). In this case, the magnitude of the fluctuations (standard deviation) is assumed to be piecewise constant before and after the change-point, representing two additional high-level parameters. Thus, in the alternative model, Poisson processes with Gaussian rate fluctuations[10,31–33] are combined with an additional change-point.

To solve the inference problem, we systematically vary the change-point and the two standard deviations (both are zero in the classical approach), and fit the distribution of the accident rate and the corresponding mean value to the data (Fig. 2a, b). For each combination of change-point and standard deviation, we obtain a different model evidence value. When multiplied with appropriate prior probability values (Supplementary Figs. 2 and 3) and normalized, these model evidence values correspond to the distribution of the change-point (Fig. 2a, b) and of the two standard deviations (Fig. 2c, d). The distributions characterize the most likely value and the uncertainty of the high-level parameter estimates. When the model evidence values are not normalized but integrated over all high-level parameter combinations, we obtain the compound model evidence which characterizes the goodness-of-fit for the two competing models (Fig. 2e).

We find that the alternative model fits the data with a 2-fold higher compound model evidence compared to the classical model and also shows a broader change-point distribution with several distinct peaks (Fig. 2b; Supplementary Fig. 4) that line up

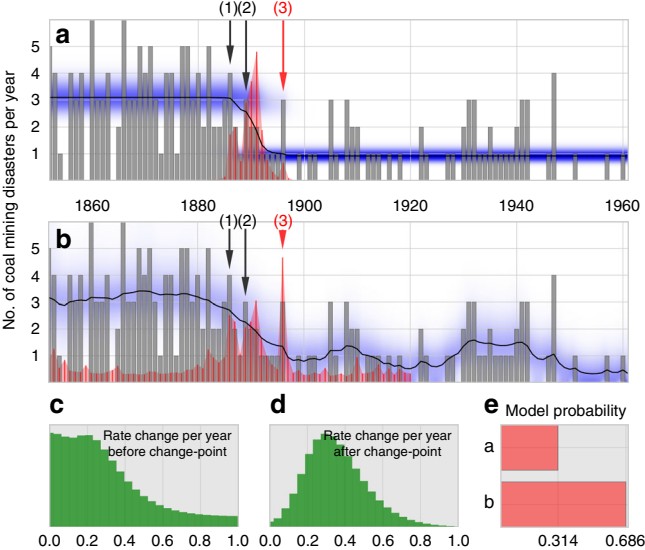

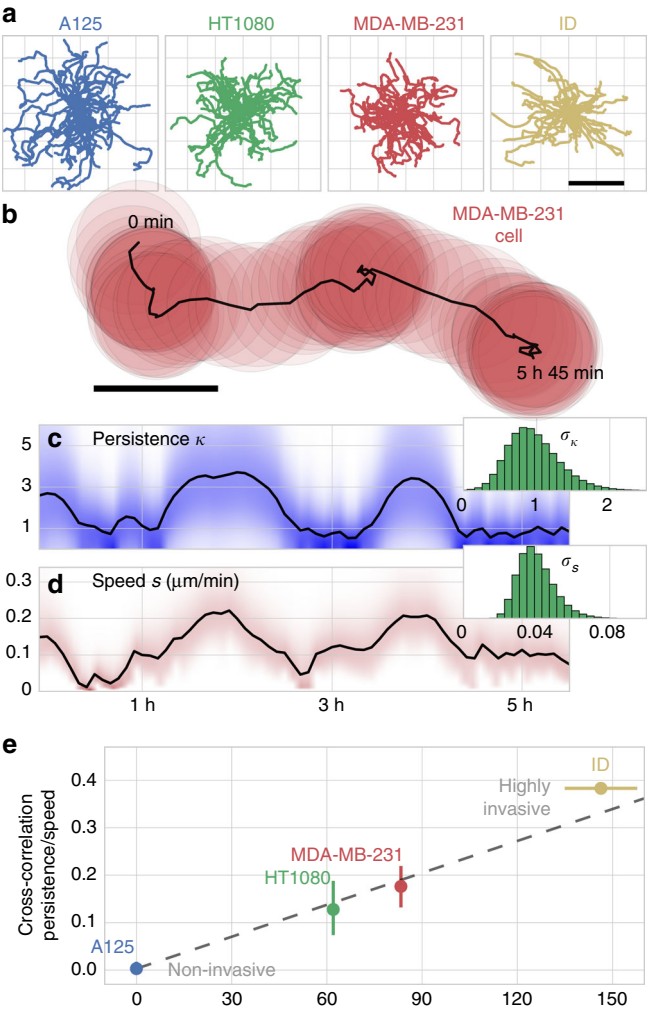

**Fig. 2** Frequency of mining accidents. **a** Reported annual number of accidents (gray bars) and inferred mean accident rate (black line), based on a classic change-point model. Blue shading indicates the distribution of the accident rate. Red bars indicate the inferred change-point distribution. Arrows indicate relevant historical events. (1) Safety report of a Royal Commission, (2) Foundation of the Miner's Federation, (3) Enactment of safety regulations. **b** Same as in **a** for the alternative model with additional Gaussian fluctuations of the accident rate. **c**, **d** Distributions of the magnitude of accident-rate fluctuations (standard deviation) before **c** and after **d** the change-point, for the alternative model shown in **b**. **e** Relative model evidence values of the classic **a** versus the alternative **b** change-point model

with relevant historical events: The first major peak occurred in 1886 and coincides with the publication of a report by the Royal Commission on Accidents in Mines[28,34]. The second peak occurred in 1891, two years after the foundation of the Miner's Federation[35]. The third peak occurred 1896 and coincides with the enactment of the Explosives in Coal Mines Order and the Coal Mines Regulation Act which finally enforced regulations such as the use of safety lamps and safer explosives[28,34]. This important transforming event marks the most plausible turning point for the reduction in mining accidents and accordingly attains the largest change-point probability in the alternative model but not in the classical model, which appears to emphasize more the foundation of the Miner's Federation (Fig. 2a).

**Tumor cell invasiveness**. The ability of tumor cells to invade interstitial tissue represents a crucial factor in the pathological process of metastasis. A common way to characterize cell motility, as a substitute for cell invasiveness, are migration experiments on flat surfaces such as a Petri dish. Unfortunately, cell motility parameters measured on 2D substrates, such as average speed or directional persistence, do not serve as a viable indicator for cell migration in 3D surrogate tissue[36–39]. This may be because 3D invasiveness is not so much a function of the average speed and persistence, as even highly invasive cancer cells usually display frequent changes between phases with low and high migratory activity[40]. Rather, tissue invasion requires phases with simultaneously high speed and high persistence. To test whether such a correlation between speed and persistence exists also for 2D migration, we analyze cell trajectories of three differently invasive cell lines (A125 lung carcinoma, MDA-MB231 breast carcinoma, HT1080 fibrosarcoma) and of highly invasive primary inflammatory duct (ID) breast carcinoma cells (Fig. 3a, b;

**Fig. 3** Heterogeneity in tumor cell migration. **a** Migration paths of 50 trajectories per cell line over the course of five hours (scale bar=50 µm). **b** Migration trajectory of an MDA-MB-231 cell on a fibronectin-coated plastic surface. The red circles indicate the nucleus size of the cell (scale bar: 10 µm). **c** Inferred mean values (black line) and distribution (shaded blue) of the time-varying directional persistence of the cell. Inset: High-level parameter distribution of the inferred standard deviation of changes in persistence per time step. **d** Inferred mean values (black line) and distribution (shaded red) of the time-varying cell speed. Inset: High-level parameter distribution of the inferred standard deviation of changes in cell speed per time step. **e** Correlation coefficient of 2D directional persistence and cell speed versus characteristic 3D invasion depth (after 3 days) of four differently invasive cell lines. The dashed line indicates the best linear fit of the data. Error bars correspond to 1 s.e.m.

Supplementary Data 2). Individual cells are continuously tracked for 2–16 h. Cell trajectories are then modeled by a random walk characterized by a step size and a turning angle. The step sizes follow a Rayleigh distribution, and the turning angles follow a Gaussian distribution confined to the unit circle (a von-Mises distribution), centered around zero. The width of the von-Mises distribution quantifies the directional persistence of a cell, and the mode of the Rayleigh distribution is a measure of average cell speed. Both of these low-level parameters (cell speed and directional persistence) are allowed to change over time, each according to a Gaussian random walk (Fig. 3c, d). The two standard deviations of this Gaussian random walk are determined separately for each individual cell (Fig. 3c, d, insets).

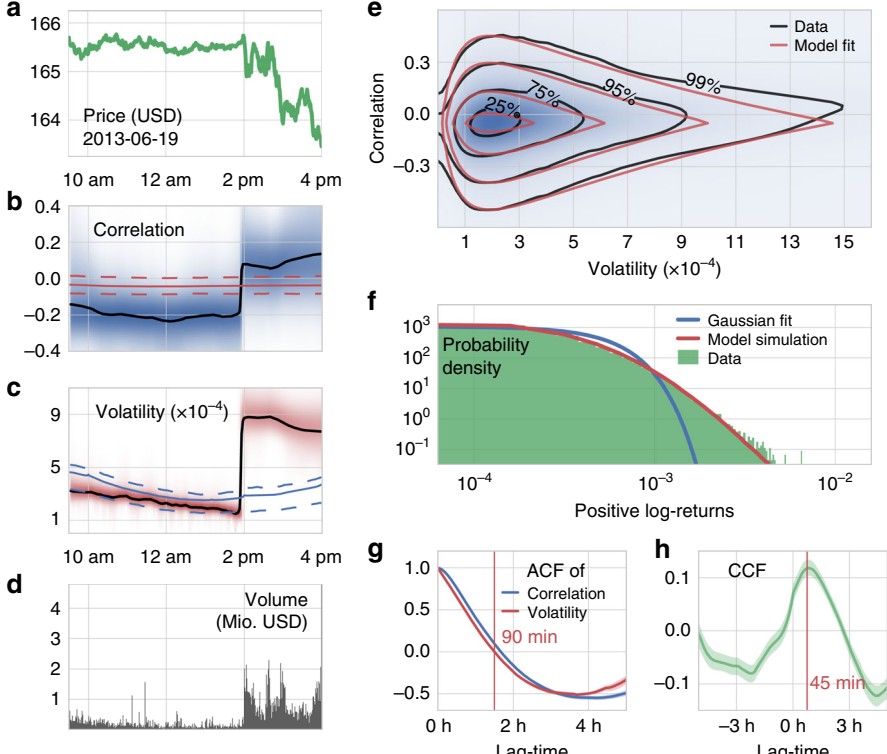

**Fig. 4** Heterogeneity in intra-day price fluctuations. **a** Price evolution of SPY on June 19th, 2013. **b** Inferred temporal evolution of the mean correlation (black line) and the corresponding parameter distribution (blue shading). Red line and dashed lines represent the average daily temporal evolution of the mean correlation and its inter-quartile range, respectively. **c** Inferred temporal evolution of the mean volatility (black line) and the corresponding parameter distribution (red shading). Blue line and dashed lines represent the average daily temporal evolution of the mean volatility and its inter-quartile range, respectively. **d** Minute-scale trading volume of SPY. **e** Joint distribution of correlation and volatility, based on the heterogeneous random walk model (black lines; blue shading) and the corresponding fit using analytical distributions (red lines). **f** Histogram of positive minute log-returns from the 2011 to 2015 of the exchange-traded fund SPY (green), together with the best fit Gaussian distribution of the data (blue line) and the simulated distribution of log-returns from the heterogeneous model (red line). **g** Auto-correlation functions of the two model parameters, correlation (blue) and volatility (red). **h** Parameter cross-correlation as a function of lag-time. Positive lag-times correspond to a comparison of future volatility values to current correlation values. The shading in **g**, **h** corresponds to 1 s.e.m.

During 2D migration, all four cell types show phases with high and low directional persistence and cell speed. Directional persistence and speed are positively correlated in three of the investigated cell types to a varying degree (Fig. 3e; Supplementary Fig. 5). By contrast, the non-invasive A125 cells show no such correlation. Furthermore, we have measured the characteristic invasion depth for the same four cell lines in a reconstituted collagen matrix (a surrogate for interstitial tissue)[41].

We find a strong linear relationship between the invasion depth in 3D matrices and the correlation strength between 2D speed and persistence (Fig. 3e), indicating that migratory phases of simultaneously high persistence and cell speed are crucial for the invasion process of tumor cells. This finding is in line with earlier work that revealed the presence of such highly efficient migratory phases for a single invasive cell line (MDA-MB-231) in 3D surrogate tissue[40]. That these migratory phases can also be detected in a 2D motility assay suggests that some aspects of 2D motility may be indicative of 3D invasive behavior, underlining the role of heterogeneity in cancer on a cellular level[42,43].

**Stock market fluctuations.** For many financial assets, the distribution of (logarithmic) returns has a "fat" tail, rendering large price fluctuations much more probable compared to a standard Gaussian random walk. Numerous approaches exist to model fat-tailed distributions of stock market fluctuations, such as the extreme value theory[44], Autoregressive Conditional

Heteroskedasticity models[45] and optimal-trade models for large market participants[46].

Here, we describe stock market returns as a correlated random walk with two parameters, volatility and correlation. Volatility measures the magnitude of price fluctuations, analogous to cell speed, while the correlation coefficient of subsequent returns quantifies the directional persistence of stock market trends. We show that fat-tailed distributions in stock market returns emerge naturally from temporal fluctuations of both volatility and correlation. These fluctuations are described by a high-level model that accounts for both, gradual and abrupt changes. Abrupt changes are implemented by assigning a minimal probability for all possible parameter values.

We analyze minute-scale fluctuations in the price of the exchange-traded fund SPY, which is specifically designed to track the value of the Standard&Poor's 500 index. On long time-scales, its value therefore reflects the current macro-economic state of the U.S. economy. Due to its large daily trading volume, the SPY moreover reflects the market micro-structure on the minute time-scale. Using our method, we obtain a minute-scale time series of volatility and correlation for each of the 1246 regular trading days from 2011 to 2015.

As an example, we show the price fluctuations of the SPY during an individual trading day (Fig. 4a) and the associated fluctuations in correlation (Fig. 4b) and volatility (Fig. 4c). On that particular day, the price was fairly constant until around 2:00

p.m. when the Federal Open Market Committee announced its latest report. After the announcement, the trading volume (Fig. 4d) increased drastically and the price began to fluctuate and to decline. Accordingly, the correlation switched from a negative to a positive value, and volatility increased by several fold. Averaged over several years, we find that a usual trading day starts with above-average volatility during the first trading minutes and then settles to a low value, only to increase towards the end of the trading day (Fig. 4c).

When we plot the volatility of all trading minutes during the years 2011–2015 versus the correlation, we obtain the joint distribution of both parameters (Fig. 4e). Accordingly, most of the time the stock market shows low volatility and a slightly negative correlation. The correlation of returns approximates a two-sided exponential (Laplace) distribution, indicating that series of strongly positive or negative correlated price fluctuations are highly unlikely. A close-to-zero correlation characterizes an efficient market, as strong correlations would render price movements predictable and could be immediately exploited.

The volatility follows a so-called compound gamma distribution[47], according to which trading days with low volatility are more likely than days with high volatility, but trading days with extremely high volatility ("Black Fridays") are still possible. The compound gamma distribution has been previously observed for minute-scale trading volumes of NASDAQ stocks[48], which underscores the known tight relationship between volatility and trading volume. Simulated price series using the two-sided exponential distribution for correlation and the compound gamma distribution for volatility accurately reproduce the fat tail seen in the data (Fig. 4f).

We further find that changes in correlation and volatility become uncorrelated after 90 min (Fig. 4g), while stock price fluctuations become uncorrelated within few minutes (Supplementary Fig. 6). Thus, a triggering event (like an unexpected news announcement) not only changes correlation and volatility momentarily, but alters the market dynamics for a period of 90 min on average. Moreover, we find a weak but significant peak in the cross-correlation function, revealing that changes in volatility tend to follow changes in correlation with a delay of 45 min (Fig. 4h). We speculate that trading strategies such as "technical analysis", which try to identify trends (periods of high correlation) in price series, trigger a higher trading activity and thus increase volatility. Although the fat-tailed distribution of returns can readily be described solely by a time-varying volatility parameter[6,49–51] (Supplementary Fig. 7), our example shows that additional insights into financial markets can be gained by investigating the connection between different market parameters.

**Real-time model selection.** Abrupt changes in market dynamics (as shown in Fig. 4b, c) can easily be detected with hindsight, taking into account all data points of a trading day (including data points generated after the parameter jump). For applications in finance, however, one is interested in detecting abrupt changes in market dynamics in real-time, as these events are often tied to the release of new, market-relevant information. If new information drastically changes the way traders act on the market, previous information about parameters like volatility and correlation will become useless.

Our method can be used to evaluate exactly this probability of current information becoming useless. In particular, we test whether each minute-to-minute price change is best described by previously obtained estimates of volatility and correlation, or by discarding previous parameter estimates (Fig. 5a). In the former case ("normal" market dynamics), the transformation that

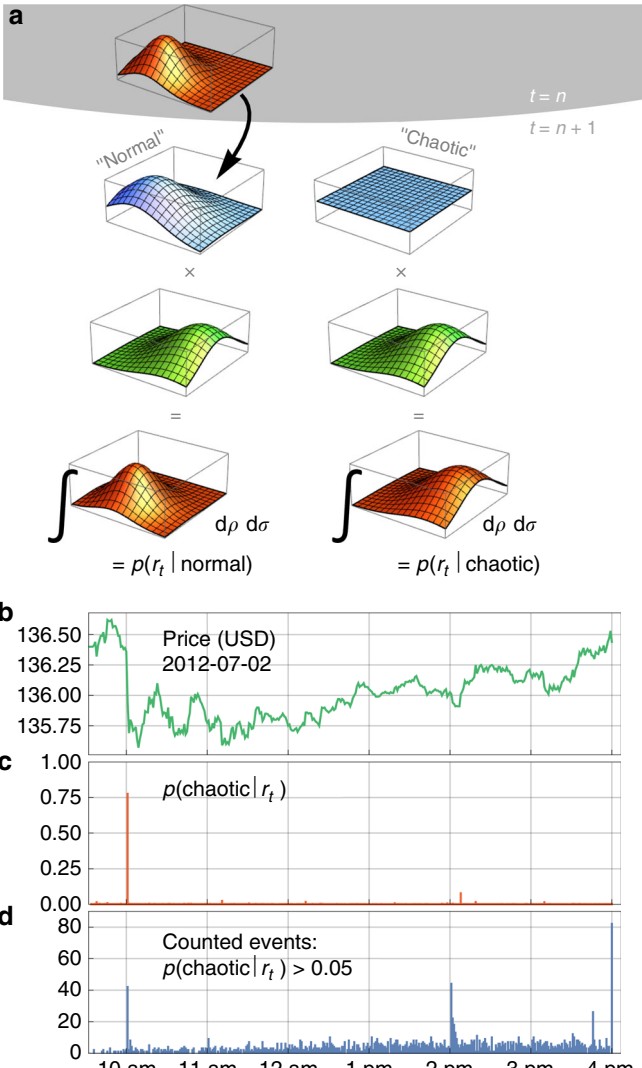

**Fig. 5** Real-time model selection. **a** Schematic illustration of a single iteration step for online model selection. To model normal market function (left column), the current posterior distribution (upper red dist.) is used to derive a new prior distribution (blue). To model anomalous price movements (right column), a flat prior distribution (blue) is assumed. Both priors are subsequently multiplied with the likelihood function (green) of a new data point to yield the (non-normalized) posterior distributions (lower red distributions). Integrating those posteriors with respect to the parameters correlation and volatility yields the corresponding model evidence values. **b** Price evolution of SPY on July 2nd, 2012. **c** Probability that previous parameter information has become useless (i.e., the probability in favor of the memoryless high-level model) for all trading minutes of SPY on July 2nd, 2012. **d** Distribution of events with a probability greater than 5% in favor of the memoryless high-level model

converts the posterior to the new prior distribution is a Gaussian convolution so that the new prior is firmly based on previous information and allows only for gradual parameter variations. In the latter case ("chaotic" market dynamics), we assume a flat prior that erases any previous parameter information. In case of abrupt changes in market dynamics, the "normal" model will fail to adapt the parameters to the new price due to its inflexible prior and thus yield a low compound model evidence, while the memoryless "chaotic" model will readily adjust the parameter estimates and thus attain a high compound model evidence.

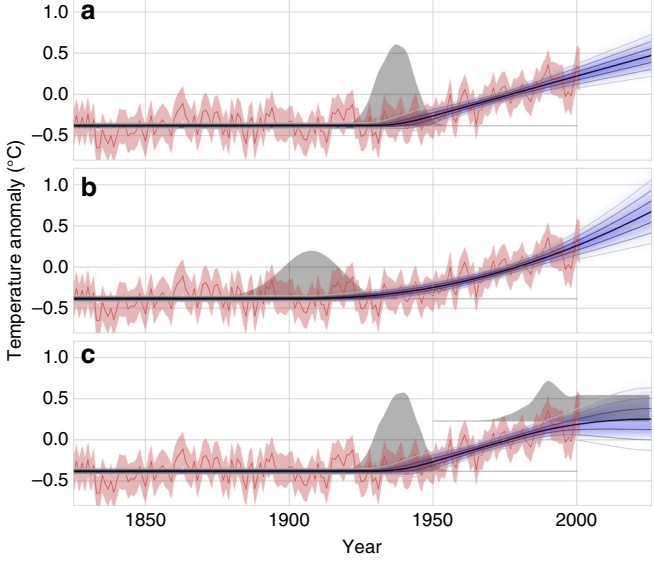

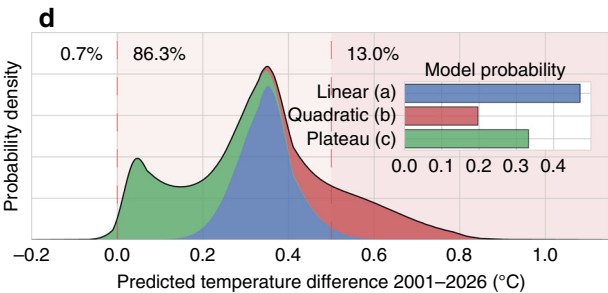

**Fig. 6** Weighting different climate change scenarios. **a** Reconstructed annual mean temperature (red line) and corresponding standard error intervals (red shading) from the PAGES 2k Consortium for the Australasia region. Assuming a linear increase in temperature after a break-point, the distribution of the inferred temperature (blue shading), together with its mean value (black line) and 1 s.d., 2 s.d., and 3 s.d.-intervals (from dark to light gray) are displayed. The inferred distribution of the breakpoint is depicted in gray. **b** Same as in **a**, but assuming a quadratic increase in temperature after a break-point. **c** Same as in **a**, but assuming a second temperature plateau after a linear warming. **d** Distribution of predicted temperature differences between 2001 and 2026, obtained by averaging of single model predictions, weighted by the respective model evidence values (inset)

Between 2011 and 2015, we detect a total of 1640 irregular events with a risk of at least 5% of rendering previous parameter estimates useless. Such events are often accompanied by large price deviations (Fig. 5b, c). Note that this risk metric is increased only at the time of the initial market perturbation, even if such an event has a prolonged effect on market dynamics (Supplementary Fig. 8).

Interestingly, the temporal distribution of such seemingly irregular events shows pronounced peaks at 10:00 a.m., 2:00 p.m., 3:45 p.m., and 4:00 p.m. (Fig. 5d). The peak at 10:00 a.m. is explained by scheduled announcements of U.S. macroeconomic indicators by the Bureau of Census (Fig. 5b, c shows the impact of a monthly announcement by the Bureau of Census on the price of SPY), the National Association of Realtors, the Conference Board and possibly others[52]. If the information that is shared by these announcements is unexpected at least to some extent, it alters market behavior and thus renders previous intra-day knowledge about market dynamics useless. In the same way, the peak at 2:00 p.m. is generated by press releases of the Federal Open Market Committee. The final two peaks occur 15 min before market close

and in the last trading minute of the day, respectively, and are associated with a higher trading activity by market participants who close their open positions at the end of the trading day to protect themselves against new information that might become available after trading hours. This temporal clustering of trading-induced anomalies is not exclusive to financial assets but instead represents a general property of exchange-traded goods and has also been reported for electrical power prices[53].

In addition to this novel risk metric introduced here, our method can be adapted to evaluate more established risk estimators such as the Value-at-Risk (Supplementary Figs. 9 and 10), providing an alternative to the commonly employed moving-window or return interval approach[4,54,55].

**Robust predictions for climate change**. As a final example, we analyze global warming[56]. Numerous models have been used to extrapolate historic climate data on regional as well as global scales and for different time ranges, from decades to centuries. Due to the wide spectrum of possible future climate scenarios, so-called multi-model projections that compute the weighted average of predictions from different climate models are thought to provide more robust climate projections[57]. However, there is no consensus on how to objectively assign weights to different models[58,59]. Here we focus on reconstructed annual temperature values of Australasia from the years 1600 to 2001, compiled by the PAGES 2k Consortium[60].

For each year, we model the temperature series using a Gaussian distribution with unknown mean as the low-level model. As high-level models for describing how the mean temperature changes on long time-scales, we implement three different climate scenarios: constant, accelerating or plateauing global warming. Although the distribution of temperature fluctuations has been described in the superstatistical context before[7], this example focuses on identifying the onset of global warming and its functional form, and further provides objective probability weights for all the three scenarios.

Our first scenario corresponds to the classic "hockey-stick graph"[61,62] and assumes a constant mean temperature beginning in the year 1600, followed by a linear increase in temperature after a breakpoint that is to be inferred from the data (Fig. 6a). We find that the transition to the linear increase occurred around 1937 (±6 years) and results in an increase of 0.010 ± 0.002 °C yr$^{-1}$. To compute the predictive distribution of future temperature values for 25 years, we supply the inference algorithm with 25 additional but empty data slots. The inference algorithm then fills these gaps, generating predictions of future parameter values. For 2026, it predicts a temperature anomaly of 0.47 ± 0.09 °C.

The second scenario is an example for accelerating climate change with a quadratic increase in temperature after a period of constant temperature (Fig. 6b). The model predicts a temperature increase starting around 1906 (±10 years), 30 years earlier than in the linear model. For 2026, it predicts a temperature anomaly of 0.68 ± 0.13 °C.

In a third model, we consider a stagnating temperature after a period of global warming. By introducing a second break-point to the linear model and assuming a constant temperature after this point, this model represents the possibility that the recent warming trend of the 20th century has come or will come to an end (Fig. 6c). The model predicts a temperature increase starting around 1938 (±6 years) and identifies 1990 as the most probable end of the warming period. For 2026, it predicts a temperature anomaly of 0.25 ± 0.13 °C.

While these models do not give us a deeper understanding of the underlying processes that drive global warming, they capture basic scenarios of climate change. All three models fit the past

temperature data reasonably well, but they arrive at diverging predictions. By computing the compound model evidence for each of the three scenarios, we find that the linear increase model is clearly favored, followed by the plateauing climate change model, while the accelerating climate change model is least likely (Fig. 6d, inset).

By averaging the predicted mean temperature distributions of all three models, weighted by their compound model evidence, we gain a robust prediction for the year 2026. We compute a chance of 0.7% that 2026 will be cooler compared to 2001. A mild temperature increase of <0.5 °C has a probability of 86.3%, leaving a 13.0% chance of a temperature increase >0.5 °C. This example demonstrates how model averaging may be used to gain robust yet easily interpretable predictions. Finally, it is important to note that we neglect any long-range correlations in the measured temperature data, because the likelihood function (low-level model) does not factorize for long-range correlated parameters, as required by our iterative inference method. While our method provides a solution to the weighting problem in multi-model climate projections, it may therefore still underestimate the uncertainty of the temperature predictions[63–65].

## Discussion

In this study, we present a novel method for the inference of time-varying parameters from noisy, short-term correlated time series data. The parameters describe the data according to a two-level hierarchical model. Importantly, our method can objectively compare different models and select the best model based on the principle of Occam's razor that weighs goodness of fit against model complexity. Inferring the parameter distributions iteratively, step by step, the computation time of our method scales linearly with the number of time steps, and in this regard outperforms Markov Chain Monte Carlo methods.

Compared to Variational Bayes techniques, our method is more easily adaptable to a large class of probabilistic and deterministic models without expert knowledge. While the representation of the model parameters on a discrete lattice limits the number of parameters, it also facilitates the direct evaluation of the model evidence and provides an objective model selection criterion. To our knowledge, no other general methods are currently available to select between competing models involving time-varying parameters.

We have applied our method to four examples from social science, cell biophysics, finance and climate research, and have demonstrated that the dynamics of the model parameters uncover relevant information about complex systems which cannot be obtained from the static mean values of the parameters alone. Furthermore, we have shown that the goodness-of-fit in change-point models can be greatly improved by additionally accounting for gradual stochastic parameter fluctuations before and after the change-point. Our method is also applicable to continuous data streams (for example sensor data in medicine, meteorology and seismology, or social data like twitter messages), thereby allowing users to compare the likelihood of different scenarios in real-time, for example normal heart function versus cardiac arrhythmia. In addition, with our method it is straight-forward to predict future parameter values and their uncertainty. In the same way, the method can bridge gaps in the measured time series.

In future work, our method could be extended to include a larger number of low-level parameters by using non-regular parameter grids. For example, a recently reported information-theoretic approach[66] maximizes the information that can be learned about the model parameters based on the available data. This method not only yields an appropriate discretization of the parameter space, but further automatically identifies parameters that are poorly constrained by the data.

To facilitate the use of this method, we have developed the open-source probabilistic programming framework bayesloop[67] written in Python (bayesloop.com).

## Methods

**Iterative evaluation of the model evidence**. In Bayesian statistics, a parameter distribution that is inferred from data based on a probabilistic model is called posterior distribution. This posterior distribution is computed as the product of a likelihood function with a prior distribution, normalized by a model evidence. Traditionally, this model evidence is computed from the integral of likelihood times prior so that the posterior distribution is properly normalized. Therefore, in the case of time series with time-varying model parameters, this integration step can only be performed after every point of the time series has been analyzed. By contrast, in our method we update the model evidence for every new data point of the time series with an iterative approach. Specifically, we have adapted the iterative approach used for evaluating Hidden Markov models[68,69] to hierarchical models that consist of a low-level model (defined by the likelihood function $L$) with time-varying parameters $\boldsymbol{\theta}_t$ ($t = 1, 2, .. N$), and a high-level model (defined by a transformation $T$) with high-level parameters $\boldsymbol{\eta}$. We discretize $\boldsymbol{\theta}$ and $\boldsymbol{\eta}$ on a regular grid, resulting in a discrete set of parameter values $\boldsymbol{\theta}^{(i)}$ with $i = 1, .. n_i$, and a set of high-level parameter values $\boldsymbol{\eta}^{(j)}$ with $j = 1, .. n_j$. Then

$$\alpha_t^{(ij)} = p\left(\{\mathbf{d}\}_{t' \le t}|\boldsymbol{\theta}_t^{(i)}\right) \cdot p\left(\boldsymbol{\theta}_t^{(i)}|\boldsymbol{\eta}^{(j)}\right) = p\left(\boldsymbol{\theta}_t^{(i)}, \{\mathbf{d}_{t'}\}_{t' \le t}|\boldsymbol{\eta}^{(j)}\right) \quad (1)$$

is the product of likelihood and prior, i.e., the non-normalized posterior (the conditional probability of the data $\mathbf{d}$ up to time step $t$ and the parameters $\boldsymbol{\theta}$ at time $t$, given the high-level parameters $\boldsymbol{\eta}$). To advance Eq. (1) by one time step, our method relies on models with a factorizable likelihood function. Each of the likelihood factors describes the probability of one data point, given the parameter values of the same time step and past data points:

$$L_{t+1}^{(i)} = p\left(\mathbf{d}_{t+1}|\boldsymbol{\theta}_{t+1}^{(i)}, \{\mathbf{d}_{t'}\}_{t' \le t}\right) \quad (2)$$

The class of models which supports such a likelihood factorization includes not only models with independent observations (e.g., a Poisson process), but also auto-regressive models for which the current data point depends on past data points (e.g., a correlated random walk model). Not supported are models in which the current data point also depends on past parameter values, as in moving-average models or models with long-range correlated data. If the likelihood function of the model is factorizable, we may advance Eq. (1) by one time step as follows:

$$\alpha_{t+1}^{(ij)} = L_{t+1}^{(i)} \cdot T_{t+1}^{(j)}\left(\alpha_t^{(ij)}\right). \quad (3)$$

Here, $T$ denotes the high-level model. It is a norm-preserving transformation as specified by the high-level parameters. $T$ defines the temporal variations of the low-level parameters and may itself depend on time. In essence, $T$ modifies the non-normalized posterior $\alpha^{(ij)}$. Norm-preserving in this context means that the transformation does not change the integral value of $\alpha$ with respect to the parameters $\boldsymbol{\theta}$. One example for a possible transformation $T$ is a convolution of $\alpha$ with a Gaussian, which allows the low-level parameters $\boldsymbol{\theta}$ to slowly change over time. Another example is the addition of $\alpha$ with a constant small value, followed by a norm-preserving multiplication with another constant value, which allows the low-level parameters $\boldsymbol{\theta}$ to suddenly change within the next time step.

Starting with an initial prior distribution $p(\boldsymbol{\theta}_0^{(i)})$ that summarizes the prior knowledge about the parameter values before taking into account any data, the iteration described in Eq. (3) is applied to all time steps. From the unnormalized posterior of the last time step $\alpha_N^{(ij)}$, we finally compute the model evidence (i.e., the probability of all data points given the hyper parameter values $\boldsymbol{\eta}^{(j)}$) by marginalizing $\alpha_N^{(ij)}$ with respect to all low-level parameters:

$$p\left(\{\mathbf{d}_{t'}\}_{t' \le N}|\boldsymbol{\eta}^{(j)}\right) = \sum_i \alpha_N^{(ij)} \cdot \Delta_{\boldsymbol{\theta}} \quad (4)$$

Here, $\Delta_{\boldsymbol{\theta}}$ represents the voxel size of the parameter grid. By maximizing the model evidence, one can optimize high-level parameter values, and at the same time select between different choices of low-level and high-level models. Note that the model evidence can be computed without inferring the time-varying parameters $\boldsymbol{\theta}^{(i)}$.

**Inference of time-varying parameters**. Once we know the model evidence, we can infer the joint parameter distribution for each time step, given past and future

data points and the high-level parameter values:

$$\mathrm{p}\left(\boldsymbol{\theta}_t^{(i)} | \{\mathbf{d}_{t'}\}_{t' \le N}, \boldsymbol{\eta}^{(j)}\right) = \frac{\alpha_t^{(ij)} \cdot \beta_t^{(ij)}}{\mathrm{p}\left(\{\mathbf{d}_{t'}\}_{t' \le N} | \boldsymbol{\eta}^{(j)}\right)}, \tag{5}$$

The non-normalized posterior $\alpha_t^{(ij)}$ in the "forward" time direction is computed from Eq. (1). $\beta_t^{(ij)}$ is the probability of all future data points, given the current (hyper-) parameter values and the current and past data points:

$$\beta_t^{(ij)} = \mathrm{p}\left(\{\mathbf{d}_{t'}\}_{t' > t} | \boldsymbol{\theta}_t^{(i)}, \mathbf{d}_{t' \le t}, \boldsymbol{\eta}^{(j)}\right) \tag{6}$$

When we restrict ourselves to low-level models with a likelihood function that factorizes as in Eq. (2), however, past observations $\{\mathbf{d}_{t'}\}_{t' < t}$ must be independent of future observations $\{\mathbf{d}_{t'}\}_{t' > t}$, and can only depend on the current parameters $\boldsymbol{\theta}_t$ and observations $\mathbf{d}_t$[69]. Exploiting this conditional independence, $\beta_t^{(ij)}$ is simplified to

$$\beta_t^{(ij)} = \mathrm{p}\left(\{\mathbf{d}_{t'}\}_{t' > t} | \boldsymbol{\theta}_t^{(i)}, \mathbf{d}_t, \boldsymbol{\eta}^{(j)}\right) \tag{7}$$

and can be efficiently computed in an iterative way, moving backwards in time:

$$\beta_t^{(ij)} = T'^{(ij)}_{t+1}\left(L_{t+1}^{(i)} \cdot \beta_{t+1}^{(ij)}\right) \tag{8}$$

The transformation $T'$ is used to incorporate the parameter dynamics in negative direction of time. If the parameter dynamics are reversible (e.g., for a Gaussian Random Walk), $T' = T$. To model non-reversible dynamics, such as deterministic trend, two separate transformations are needed.

**Model averaging**. For most high-level models, the optimal high-level parameter values are not known a-priori, for example the magnitude of volatility fluctuations in the case of stock market models. Therefore, one may choose a discrete set of high-level parameter values $\boldsymbol{\eta}^{(j)}$ that cover an interval of interest, and then run the inference algorithm described above for each individual high-level parameter value $\boldsymbol{\eta}^{(j)}$, each time resulting in a model evidence $\mathrm{p}\left(\{\mathbf{d}_t\}_{t' \le N} | \boldsymbol{\eta}^{(j)}\right)$. From these individual runs, we compute the compound model evidence which takes into account the uncertainty of the high-level parameters:

$$\mathrm{p}\left(\{\mathbf{d}_t\}_{t \le N}\right) = \sum_j \mathrm{p}\left(\{\mathbf{d}_t\}_{t \le N} | \boldsymbol{\eta}^{(j)}\right) \cdot \mathrm{p}\left(\boldsymbol{\eta}^{(j)}\right) \tag{9}$$

Here, $\mathrm{p}(\boldsymbol{\eta}^{(j)})$ denotes the prior distribution of the high-level parameters. If not specified otherwise, we assign equal probability to all $\boldsymbol{\eta}^{(j)}$. Using Eq. (9), the joint distribution of the high-level parameters can be determined:

$$\mathrm{p}\left(\boldsymbol{\eta}^{(j)} | \{\mathbf{d}_t\}_{t \le N}\right) = \frac{\mathrm{p}\left(\{\mathbf{d}_t\}_{t \le N} | \boldsymbol{\eta}^{(j)}\right) \cdot \mathrm{p}(\boldsymbol{\eta}^{(j)})}{\mathrm{p}\left(\{\mathbf{d}_t\}_{t \le N}\right)} \tag{10}$$

In essence, this is the normalized model evidence of each individual run with fixed high-level parameters. This high-level parameter distribution can now be used to compute the time course of the weighted average distribution of the low-level parameters from each individual run of the inference algorithm:

$$\mathrm{p}\left(\boldsymbol{\theta}_t^{(i)} | \{\mathbf{d}_{t'}\}_{t' \le N}\right) = \sum_j \mathrm{p}\left(\boldsymbol{\theta}_t^{(i)} | \{\mathbf{d}_{t'}\}_{t' \le N}, \boldsymbol{\eta}^{(j)}\right) \cdot \mathrm{p}\left(\boldsymbol{\eta}^{(j)} | \{\mathbf{d}_t\}_{t \le N}\right) \tag{11}$$

**Prediction and handling of missing data**. To infer the parameter distribution of time steps for which no data are available, either because data is missing or because the time steps lie in the future, we may simply replace the likelihood function of the low-level model by a flat, improper (non-normalized) distribution:

$$L_t^{(i)} = 1 \ \forall i \tag{12}$$

Inserting this into Eqs. (3) and (8), we find that the iteration from $\alpha_t^{(ij)}$ to $\alpha_{t+1}^{(ij)}$ (and from $\beta_{t+1}^{(ij)}$ to $\beta_t^{(ij)}$) is given by transforming the current distributions according to the high-level model, without adding any information from data.

**Online model selection**. Assume that we have (for simplicity) two mutually exclusive high-level models $A$ and $B$ with high-level parameter values $\mathbf{a}^{(j)}$ and $\mathbf{b}^{(k)}$, e.g., one for gradual parameter changes and one for abrupt parameter jumps. In an analysis of an on-going data stream, we want to compute the relative probability that each of those two high-level models describe only the latest data point $t_{\mathrm{now}}$. First, we compute the hyper-model evidence of only the latest data point, using Eq. (9), by dividing the model evidence after seeing the data point of time step $t_{\mathrm{now}}$ by

the model evidence before seeing this data point:

$$\mathrm{p}\left(\mathbf{d}_{t_{\mathrm{now}}} | A, \{\mathbf{d}_t\}_{t \le (t_{\mathrm{now}} - 1)}\right) = \frac{\mathrm{p}\left(\{\mathbf{d}_t\}_{t \le t_{\mathrm{now}}} | A\right)}{\mathrm{p}\left(\{\mathbf{d}_t\}_{t \le (t_{\mathrm{now}} - 1)} | A\right)}, \tag{13}$$

and analogous for high-level model $B$.

For independent observations, i.e., if the probability of the current observation does not depend on past observations given the current parameter values (as it is the case for all examples given in this report except for the auto-regressive model of stock market fluctuations), the expression above simplifies to:

$$\mathrm{p}\left(\mathbf{d}_{t_{\mathrm{now}}} | A, \{\mathbf{d}_t\}_{t \le (t_{\mathrm{now}} - 1)}\right) = \mathrm{p}\left(\mathbf{d}_{t_{\mathrm{now}}} | A\right) \tag{14}$$

Using a prior probability $\mathrm{p}(A)$ and $\mathrm{p}(B)$ to state our initial belief that the data point is described by either model $A$ or model $B$ ($\mathrm{p}(A) + \mathrm{p}(B) = 1$), we compute the relative probabilities that each of the two high-level models describes the current data point:

$$\mathrm{p}\left(A | \mathbf{d}_{t_{\mathrm{now}}}\right) = \frac{\mathrm{p}\left(\mathbf{d}_{t_{\mathrm{now}}} | A\right) \cdot \mathrm{p}(A)}{\mathrm{p}\left(\mathbf{d}_{t_{\mathrm{now}}} | A\right) \cdot \mathrm{p}(A) + \mathrm{p}\left(\mathbf{d}_{t_{\mathrm{now}}} | B\right) \cdot \mathrm{p}(B)}, \tag{15}$$

and analogous for high-level model $B$.

If the current observation depends on the previous observation (as in our stock market example where we use an autoregressive model of first order), Eq. (13) simplifies to:

$$\mathrm{p}\left(\mathbf{d}_{t_{\mathrm{now}}} | A, \{\mathbf{d}_t\}_{t \le (t_{\mathrm{now}} - 1)}\right) = \mathrm{p}\left(\mathbf{d}_{t_{\mathrm{now}}} | A, \mathbf{d}_{(t_{\mathrm{now}} - 1)}\right) \tag{16}$$

In this case, we can compute the relative probability that each of the two high-level models describe the current and the previous data point:

$$\mathrm{p}\left(A | \mathbf{d}_{t_{\mathrm{now}}}, \mathbf{d}_{(t_{\mathrm{now}} - 1)}\right)$$
$$= \frac{\mathrm{p}\left(\mathbf{d}_{t_{\mathrm{now}}} | A, \mathbf{d}_{(t_{\mathrm{now}} - 1)}\right) \cdot \mathrm{p}\left(A | \mathbf{d}_{(t_{\mathrm{now}} - 1)}\right)}{\mathrm{p}\left(\mathbf{d}_{t_{\mathrm{now}}} | A, \mathbf{d}_{(t_{\mathrm{now}} - 1)}\right) \cdot \mathrm{p}\left(A | \mathbf{d}_{(t_{\mathrm{now}} - 1)}\right) + \mathrm{p}\left(\mathbf{d}_{t_{\mathrm{now}}} | B, \mathbf{d}_{(t_{\mathrm{now}} - 1)}\right) \cdot \mathrm{p}\left(B | \mathbf{d}_{(t_{\mathrm{now}} - 1)}\right)}. \tag{17}$$

In our stock market example, we set fixed prior probabilities for each high-level model $\mathrm{p}\left(A | \mathbf{d}_{(t_{\mathrm{now}} - 1)}\right) = \mathrm{p}(A)$ in each time step, and analogous for $B$.

**Cell culture and experiments**. The data for the 2D migration assay and the 3D invasion assay analyzed in this work are part of a larger dataset that is described in detail in ref. [70]. All reagents were obtained from Gibco unless stated otherwise. All cell lines are maintained at 37 °C, 5% $CO_2$ and 95% humidity. MDA-MB-231 cells (obtained from the American Type Culture Collection (ATCC)) and A125 cells (gift from Peter Altevogt) are cultured in low glucose (1 g L$^{-1}$) Dulbecco's modified Eagle's medium supplemented with 10% fetal calf serum, 2 mM L-glutamine, and 100 U ml$^{-1}$ penicilin-streptomycin. HT1080 cells (obtained from the ATCC) are cultured in advanced Dulbecco's modified Eagle's medium F-12 and supplemented with 5% fetal calf serum, 2 mM L-glutamine, and 100 U ml$^{-1}$ penicillin-streptomycin. Primary cells isolated from a patient with inflammatory duct (ID) breast cancer are maintained in collagen-coated dishes in Epilcut-C medium (Stem Cell Technologies), supplemented with 1× Supplement C, 5% fetal calf serum, 2 mM L-glutamine, 50 U ml$^{-1}$ penicillin-streptomycin and 0.5 mg ml$^{-1}$ hydro-cortisone (Stem Cell Technologies). Before plating, cells are rinsed with PBS and detached with 0.05% Trypsin-ethylenediaminetetraacetic acid (Trypsin-EDTA).

For the 2D cell migration assay, we use fibronectin-coated petri dishes. Cells are plated 24 h prior to the beginning of the measurement. Cells are subsequently imaged every 5 min for 24 h. For the statistical analysis, we select cells that were tracked continuously for at least 2 h and migrated at least 30 μm away from their original position (Supplementary Data 2).

To assess the invasiveness of the cells in a 3D environment, collagen invasion assays are performed as described in ref. [41]. Cells are seeded on the surface of a 1 mm thick collagen gel. After a 3-day incubation period, the invasion depth of each cell is determined from the z-position of the stained cell nuclei. The characteristic invasion depth is computed by modeling the cumulative probability of finding a cell below a given depth as an exponential function.

**Random walk model of cell migration**. To analyze the directional persistence of individual cell migration paths, we first compute the turning angle $\phi$ between two subsequent cell movements $\mathbf{v}$ (vectorial difference of positions $\mathbf{v_t} = \mathbf{r_t} - \mathbf{r}_{t-1}$):

$$\phi_t = \mathrm{atan2}\left(v_{yt} - v_{y(t-1)}, v_{xt} - v_{x(t-1)}\right) \tag{18}$$

where atan2($y$, $x$) denotes the multi-valued inverse tangent function. It is defined as

$$
\text{atan2}(y, x) = \begin{cases} \arctan(y/x) & \text{if } x>0 \\ \arctan(y/x) + \pi & \text{if } x<0 \text{ and } y \geq 0 \\ \arctan(y/x) - \pi & \text{if } x<0 \text{ and } y<0 \\ +\pi/2 & \text{if } x = 0 \text{ and } y>0 \\ -\pi/2 & \text{if } x = 0 \text{ and } y<0 \\ \text{not defined} & \text{if } x = 0 \text{ and } y = 0 \end{cases} \quad (19)
$$

To model the measured series of turning angles, we choose a von-Mises distribution that is centered around zero as the low-level model (likelihood function):

$$
L_t^{(i)} = \text{p}\left(\phi_t | \kappa_t^{(i)}\right) = \frac{e^{\kappa_t^{(i)} \cos(\phi_t)}}{2\pi \mathbf{I}_0\left(\kappa_t^{(i)}\right)} \quad (20)
$$

where $\mathbf{I}_0$ denotes the modified Bessel function of order zero and the parameter $\kappa$ indicates a cell's directional persistence, from $\kappa = 0$ for a diffusive random walk to ballistic motion as $\kappa \to \infty$. For the inference algorithm, we use a discrete set of $n_i = 1000$ equidistant values for $\kappa$ in the interval $\kappa^{(i)} \in {}]0, 20[$. We further use a flat prior distribution for $\kappa$ to initialize the inference algorithm.

The high-level transformation $T$ is given by a discrete convolution with a Gaussian kernel and depends on a single high-level parameter, the standard deviation $\sigma$ of this kernel. To cover a wide range of parameter dynamics, from constant persistence over gradual changes to very fast changes in persistence, we choose a large high-level parameter space for the discretized high-level parameter $\sigma^{(j)} \in [0, 5]$ with $n_j = 50$ equidistant values.

To model the measured series of cell speed values $v_t = |\mathbf{v}_t|$, we choose a Rayleigh distribution as the low-level model (likelihood function):

$$
L_t^{(i)} = \text{p}\left(v_t | s_t^{(i)}\right) = \frac{v_t \cdot e^{-v_t^2/(2s_t^{(i)2})}}{s_t^{(i)2}} \quad (21)
$$

with the mode parameter $s$ indicating the most probable cell speed. For discretization, we choose $n_i = 1000$ and $s^{(i)} \in {}]0, 1.5[$ μm min$^{-1}$. To initialize the inference algorithm, we use the non-informative Jeffreys prior of the Rayleigh distribution: $\text{p}(s) \propto 1/s$. Again, a convolution with a Gaussian kernel is used as the high-level transformation $T$, and the high-level parameter space is chosen as follows: $\sigma^{(j)} \in [0, 0.2]$ μm min$^{-1}$ with $n_j = 50$.

To compute the correlation coefficient between persistence and cell speed, we use the mean values of the parameter distributions (see Eq. (11)):

$$
\overline{\kappa}_t = \sum_i \kappa^{(i)} \cdot \text{p}\left(\kappa_t^{(i)} | \{\phi_{t'}\}_{t' \leq N}\right), \quad (22)
$$

and analogous for the cell speed $s_t$. Finally, we compute the correlation coefficient $\rho$ of $\overline{\kappa}_{tk}$ and $\overline{s}_{tk}$, for all times $t$ and all cells $k$:

$$
\rho = \frac{\left\langle \left(\overline{\kappa}_{tk} - \mu_\kappa\right)\left(\overline{s}_{tk} - \mu_s\right)\right\rangle_{tk}}{\sigma_\kappa \cdot \sigma_s} \quad (23)
$$

with $\mu_\kappa$, $\mu_s$ denoting the sample mean, and $\sigma_\kappa$, $\sigma_s$ denoting the sample standard deviation of all times and cells. Finally, as our method does not provide error estimates for $\rho$, we verify the inference accuracy of individual cell trajectories by bootstrapping, see Supplementary Fig. 5.

**Minute-scale financial data**. Pricing and volume data of the exchange-traded fund SPY were accessed via the hosted research environment of the algorithmic trading platform Quantopian.com. The analysis uses closing prices of all trading minutes on regular trading days from 2011 to 2015. All prices are based on raw pricing data, without adjusting for dividends.

**Autoregressive model of price fluctuations**. Given the minute by minute closing prices $s_t$, we compute the log-returns $r_t = \log(s_t/s_{t-1})$. The time series of log-returns $r_t$ is modeled by a scaled auto-regressive process of first order (AR-1), which is defined by the following recursive instruction:

$$
r_t = \rho_t \cdot r_{t-1} + \sqrt{1 - \rho_t^2} \cdot v_t \cdot \epsilon_t \quad (24)
$$

$\rho_t$ represents the time-varying correlation coefficient of subsequent return values and is a measure of market inertia. $v_t$ denotes the time-varying standard deviation of the stochastic process and therefore represents a measure of volatility. $\epsilon_t$ is drawn from a standard normal distribution and represents the driving noise of the process. With $\overline{v}_t = \sqrt{1 - \rho_t^2} \cdot v_t$ for simplification, we obtain the following low-

level model (likelihood function) for the log-return values:

$$
L_t^{(i_\rho, i_v)} = \text{p}\left(r_t | \rho_t^{(i_\rho)}, v_t^{(i_v)}, r_{t-1}\right) = \frac{1}{\overline{v}_t^{(i_v)} \sqrt{2\pi}} \exp\left(-\frac{\left(r_t - \rho_t^{(i_\rho)} \cdot r_{t-1}\right)^2}{2\overline{v}_t^{(i_v)2}}\right) \quad (25)
$$

Note that we have two indices $i_\rho$, $i_v$ for the discretization of the joint parameter space. The discrete values $\rho_t^{(i_\rho)}$ cover the interval $]-1, 1[$, and the discrete values $v_t^{(i_v)}$ cover the interval $]0, 0.006[$. The grid dimensions are $100 \times 400$. We use a flat prior to initialize the inference algorithm.

The temporal evolution of the two low-level parameters $\rho_t$ and $v_t$ is described by a two-part high-level transformation: to cover gradual variations, the parameters are subject to Gaussian fluctuations with standard deviations $\sigma_\rho \in [0, 15] \times 10^{-2}$ and $\sigma_v \in [0, 15] \times 10^{-5}$, with 10 and 20 equally spaced values for $\sigma_\rho$ and $\sigma_v$ respectively. To account for abrupt changes, we assign a minimal probability $p_{\min} \in \{0, 10^{-6}, 10^{-3}\}$ (with respect to the probability of a flat distribution) to all parameter values at each time step. All 600 high-level parameter combinations are assigned equal probability prior to fitting.

We finally add up the parameter distributions (marginalized with respect to the hyper-parameters, c.f. Eq. (11)) of all time steps and all trading days to get the parameter distribution shown in Fig. 4e. We approximate this distribution by a product of two independent parameter distributions: $\text{p}(\rho, v) = \text{p}(\rho) \cdot \text{p}(v)$. For $\text{p}(\rho)$, we choose a Laplace (two-sided exponential) distribution with a mean of $-0.047$ and a scale of $0.077$. For $\text{p}(v)$, we follow[48] and choose a compound gamma distribution:

$$
\text{p}(v) = \int_0^\infty G(v; \alpha, p) \cdot G(p; \beta, q) \text{d}p \quad (26)
$$

$G(x; a, b)$ denotes the Gamma distribution with shape $a$ and inverse scale $b$. The estimated parameter values are $\alpha = 10$, $\beta = 6.2$ and $q = 1.65 \times 10^{-4}$.

With this model, we simulate $10^6$ series of log-returns (each of length 100) with parameters drawn from the approximated distribution $\text{p}(\rho, v)$. The resulting histogram of log-return values of this simple model closely follow the data from the years 2011 to 2015 (Fig. 4f).

The auto-correlation and cross-correlation functions of $\rho_t$ and $v_t$ that are shown in Fig. 4g, h are computed from the mean volatility and mean correlation for each minute of each trading day, minus the daily average. The correlation functions thus analyze relative changes of of $\rho_t$ and $v_t$ during a trading day.

**Real-time model selection for price fluctuations**. To distinguish in real-time between "normal" market dynamics with only gradual variations of volatility and correlation, and "chaotic" market dynamics with vanishing market memory, we compute the compound model evidence for both of these high-level models for each trading minute. The low-level model for both cases is the scaled AR-1 process as described in the previous section. The "normal" market model assumes only Gaussian fluctuations of volatility $v_t$ and correlation coefficient $\rho_t$ (i.e., without a finite minimum probability $p_{\min}$ at extreme values). The grid values for $\sigma_\rho$ and $\sigma_v$ are chosen as in the previous section. The "chaotic" market model assigns a flat prior distribution for $v_t$ and $\rho_t$ in each time step, effectively erasing prior knowledge about the time-varying parameter values. We further a-priori assume that the "chaotic" model applies once during a regular trading day, i.e., we assign a prior probability of 389/390 to the "normal", and 1/390 to the "chaotic" model in each time step.

**Historic data on coal-mining accidents**. The dataset consists of time intervals (in days) between coal-mining accidents in the United Kingdom involving ten or more men killed. It was first discussed in ref. [25], and later corrected and extended in ref. [27]. The dataset ranges from 15 March 1851 to 22 March 1962. These time intervals are binned to give the number of accidents per year, excluding the years 1851 and 1962 due to incomplete data (Supplementary Data 1).

**Heterogeneous Poisson models for accident rates**. The annual number of accidents $k_t$ is assumed to be Poisson-distributed, resulting in the low-level model (likelihood function):

$$
L_t^{(i)} = \text{p}\left(k_t | \lambda_t^{(i)}\right) = \frac{\lambda_t^{(i)k_t} \cdot e^{-\lambda_t^{(i)}}}{k_t!} \quad (27)
$$

with the time-varying accident rate $\lambda$ that is to be inferred from data. For discretization, we choose $n_i = 1000$ and $\lambda^{(i)} \in {}]0, 6[$ yr$^{-1}$. We further use the non-informative Jeffreys prior of the Poisson distribution $\text{p}(\lambda) \propto 1/\sqrt{\lambda}$ to initialize the inference algorithm.

For the classic change-point model (Fig. 2a), we assume a single change-point at a specific time step, at which we assign the non-informative Jeffreys prior for the next time step. Before and after this change-point, the accident rate is assumed to be constant. We restrict the change-point to years before 1921. From the model

evidence values, we compute the change-point distribution shown in Fig. 2a (red bars).

In the alternative model shown in Fig. 2b, the accident rate before and after the change-point is additionally subject to Gaussian fluctuations with standard deviations $\sigma^{(j)} \in [0, 1]$ with $n_j = 25$. We infer the distributions of $\sigma$ before and after the change-point (Fig. 2c, d). We assume a flat prior distribution for $\sigma$ both before and after the change-point. The results of this analysis are found to be robust against choosing different high-level priors, see Supplementary Figs. 2 and 3.

Since this example is based on a small dataset of only 110 data points, we further verify the inference results by bootstrapping, see Supplementary Fig. 4.

**Continental-scale climate data**. The paleoclimatic temperature reconstructions used in this work are part of a larger study by the PAGES 2k Consortium[60], covering seven continental-scale regions during the past one to two millennia. We focus on annual reconstructed temperature values of the Australasia region, from 1600 to 2001. The reconstructed values are based on proxy data, in this case corals, tree rings and speleothems and are given as mean annual temperature anomalies relative to a 1961–1990 reference period together with a two-standard-error interval.

**Modeling different climate scenarios**. The annual reconstructed temperature anomalies $m_t$ are assumed to follow a Gaussian distribution with known standard deviation $\sigma_t$ (computed from the given two-standard-error intervals):

$$L_t^{(i)} = \text{p}\left(m_t | \mu_t^{(i)}, \sigma_t\right) = \frac{1}{\sqrt{2\pi}\sigma_t} \exp\left(-\frac{\left(m_t - \mu_t^{(i)}\right)^2}{2\sigma_t^2}\right) \qquad (28)$$

where $\mu_t^{(i)}$ represents the inferred mean temperature anomaly that is discretized using $n_i = 15{,}000$ equally spaced values within the interval $]-1, 4[$ °C. All three scenarios assume a constant mean temperature until a change-point $t_1$: $\mu_t = \text{const.}$ for $t \leq t_1$. We a-priori assume the change-point $t_1$ to lie between the years 1825 and 2001, with equal prior probabilities.

The first scenario models the mean temperature after the first change-point as a linear increase: $\mu_t = a \cdot t$ for $t > t_1$, with the slope $a$ as an additional high-level parameter. We discretize the slope $a$ based on 50 equally spaced values within the interval $[0, 0.02]$ °C yr$^{-1}$. We use a flat prior distribution for $a$ to initialize the inference algorithm.

The second scenario models the mean temperature after the first change-point as a quadratic increase: $\mu_t = (b \cdot t)^2$ for $t > t_1$, with the coefficient $b$ as an additional high-level parameter. We assume the change-point $t_1$ to lie between the years 1825 and 2001. We discretize the coefficient $b$ exactly like the high-level parameter $a$ in the first scenario and again use a flat prior distribution for $b$ to initialize the inference algorithm.

Finally, the third scenario models the mean temperature after the first change-point as a linear increase up to a second change-point ($\mu_t = a \cdot t$ for $t_1 < t < t_2$), and assumes a constant mean temperature thereafter. We assume the first change-point $t_1$ to lie between the years 1825 and 2001 with equal prior probability, the second between 1950 and 2026. We discretize the slope $a$ as in the first scenario.

**Code availability**. The statistical inference method introduced in this work is implemented in the Python package bayesloop[67]. The software is open source (under the MIT License) and is hosted on GitHub (https://github.com/christophmark/bayesloop). The website bayesloop.com further provides access to code examples, tutorials and documentation. bayesloop uses functionality of other Python modules, namely NumPy[71], SciPy (http://www.scipy.org), SymPy[72], Matplotlib[73], Pathos[74], and tqdm[75].

**Data availability**. The coal-mining accident data shown in Fig. 2 are provided as Supplementary Data 1. The cancer cell trajectories shown in Fig. 3 are provided as Supplementary Data 2. The financial data shown in Figs. 4 and 5 can be accessed via the hosted research environment of the algorithmic trading platform Quantopian.com. The climate data shown in Fig. 6 is included in the Supplementary Information of ref. [60]. (Database S2; https://media.nature.com/original/nature-assets/ngeo/journal/v6/n5/extref/ngeo1797-s3.xlsx).

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

## Acknowledgements

This work was supported by Deutsche Forschungsgemeinschaft grants FA-336/11-1, STR 923/6-1, and the Research Training Group 1962 "Dynamic Interactions at Biological Membranes: From Single Molecules to Tissue".

## Author contributions

C.Ma. and B.F. designed the study. C.Ma. and C.Me. developed the statistical method. C. Ma. performed the analyses. L.L., P.S., and R.S. established the primary breast cancer cell line, performed the cell experiments and image analysis. C.Ma., C.Me., and B.F. wrote the paper. All authors read and approved the final manuscript.

## Additional information

**Competing interests:** The authors declare no competing interests.

