## [Peer Review File · Nature Communications]

Reviewers' comments:

Reviewer #1 (Remarks to the Author):

The manuscript "Bayesian model selection for complex dynamical systems" by Mark et al., describes an empirical approach for estimating model parameters that gave rise to time series data when the parameters could potentially vary in time. The approach described is a two-level hierarchical model accompanied by a Bayesian scheme to estimate parameters and select a model. The proposed approach differs from a standard Bayesian updating procedure through the existence of a high-level model in the form of a transformation T . A Bayesian update is performed after each time point, rather than using the resulting posterior as the prior for the next update, the posterior is first transformed by T before becoming the new prior. If multiple models are to be considered, the authors discriminate among them using the marginalized likelihood or model evidence.

I enjoyed reading this paper. It is well-written. It addresses an important topic, and I learned something new from reading it. I was impressed with the breadth of applications the authors consider. I have some suggestions and concerns that I discuss below, but I believe this is a manuscript that could ultimately be accepted by the journal.

My major concern is that the authors do not discuss the limitations of their methods. For example, the authors represent probability distributions by gridding parameter space. This effectively limits their application to models with only a few parameters. Consequently, they are forced to deal with simple, phenomenological models of complex processes. I do not think a more complex, mechanistic (or physics-based) model (that includes hundreds or thousands of possibly time-varying parameters) could be considered. Along these lines, I would like to see a frank discussion of what types of data and models can be reasonably considered within their framework. Are there any underlying assumptions about the way parameters vary in time? For example, if the parameters were to vary with a characteristic time scale faster than data sampling frequency, would this pose any problems? What if you use the wrong transformation between time steps?

Related to the last point, I wondered why the authors chose a regular grid to represent the probability distribution? The ultimate goal is to approximate integrals, and I suspect performance would be improved by using Gauss points (i.e., quadrature).

Minor concerns:

In the opening sentences of the abstract, the authors write that seemingly chaotic dynamics are difficult to model because the model parameters vary in time. I think they mean to say that the model parameters "can vary in time." Although the difference is subtle, I would point out that the definition of "complex system" and "chaotic dynamics" is one of active, ongoing discussion. As it is currently written, it appears the authors are taking an ideological stand that this is their definition of "complex system" and "chaotic dynamics" when I believe they simply wish to introduce the specific problem they consider.

I appreciate that the authors' code is freely available on github. I would further recommend they get a dedicated DOI for the version of the code associated with this manuscript and cite it in their manuscript.

Occam's razor as a rule for model selection is mentioned several times. On this point, the authors may find work of Balasubramanian useful/interesting (Balasubramanian, Vijay. "Statistical inference, Occam's razor, and statistical mechanics on the space of probability distributions." *Neural computation* 9.2 (1997): 349-368). More recent (though unpublished) work of LaMont and Wiggins (arXiv:1706.01428) and Mattingly et al. (arXiv:1705.01166) attempt to take these ideas further.

Reviewer #2 (Remarks to the Author):

The paper provides useful methods to systematically estimate (with Bayesian approaches) time-varying parameters in complex superstatistical systems. Four examples from different areas are studied: Mining accidents, stock market returns, cell migration and global warming. Overall this is interesting and relevant research, but the paper is not referring in a sufficient way to previous work in this general area.

Basically, what the authors do is now well-established under the name "superstatistics", and many groups are working on similar topics, but these other groups are not cited at all. So I think it is essential to mention in the introduction and abstract that this paper is basically considering ***superstatistical*** complex systems, not general complex systems which can be anything. The authors are looking at systems where certain parameters that are usually kept constant fluctuate on a long time scale, and they have a simple local model that depends on these parameters, and they provide useful methods for local model validation. But one misses the most essential papers on superstatistical methods in the list of references, such as Beck and Cohen *Physica A* 322, 267 (2003), or Beck, Cohen, Swinney (*PRE* 72, 056133 (2005)) which extracts superstatistical parameter

distributions (similar as the ones considered in this paper) from a given time series. Surprisingly, in their previous work (ref [10]) the authors mention superstatistics in the title of their paper, but they completely ignore it in the current paper, for reasons unclear to me.

The authors should add a paragraph on superstatistics in their introduction and put their work into context. It is essential to cite more recent systematic papers on superstatistical dynamics, such as the important work by Chechkin et al in *Phys. Rev. X* 7, 021002 (2017) or the new work by Schaefer et al in *Nature Energy* which appeared in January 2018 and which applies it to yet another example, frequency fluctuations in power grids. Otherwise, the current manuscript completely misses out on new recent important developments in this field.

Finally, the authors should better explain what is really new in their approach, as compared to previous approaches. I do appreciate there is something new: The systematic Bayesian approach, and also the fact that they typically study 2 superstatistical parameters rather than 1, such as volatility and correlation, whereas previous papers mainly studied just the volatility (though papers such as Jizba et al., *Physica A* 493, 29 (2018) also study several superstatistical parameters in financial time series).

Concerning the four specific applications in this paper, people have worked on similar applications before, but again references to previous work are missing. The Poisson process model with varying rate as displayed in Fig. 2b has already been studied previously as a simple model for (not mining accidents but) train delay statistics, see Briggs et al *Physica A* 378, 498 (2007). Cancer cell migration has been previously discussed in a superstatistical context by Chen et al in *Physica A* 387, 3126 (2008), so when discussing the heterogeneity in Fig.3 this previous work could be cited (though the approach there is a bit different) and then it should be explained what is new as compared to ref [10], the previous work of the authors. Stock market fluctuations have been studied by many authors in a superstatistical context before, so the authors should make clearer what is new in their approach, as compared to references cited in Jizba et al *Physica A* 2018. The abrupt parameter changes in Fig. 5d, which occur at certain times of the day for share price index movements, are also seen in electricity markets, see e.g. Fig.1 of Schaefer et al in *Nature Energy* 2018, where they are caused by trading intervals. Global warming has also been studied in a superstatistical context before, see. e.g. Yalcin et al *Physica A* 392, 5431 (2013), but the approach in the current manuscript in Fig. 6 is different---the authors should better explain that they look at critical points where a drastic change happens, which is new and was not done in previous work. A minor comment: What is atan^2 in eq.(18)?

Overall I think significant revision of this paper is necessary before it can be accepted.

Reviewer #3 (Remarks to the Author):

Referee report for the ms entitled "Bayesian model selection for complex dynamic systems"

General remarks

The ms focuses on Bayesian updating approach to the optimization of the model selection in complex dynamic systems. In general, the presented study is based on a solid theoretical background in Bayesian statistics. The ms is clearly written and is suitable for a general audience. It is commendable that the authors have implemented their solutions as a software package and made it available for the research community. The authors also provide several examples of the application of their approach to real-world complex systems.

I do not see any major theoretical breakthrough that first appeared in this ms. The mathematical background for optimal filtering, interpolation and prediction problems including non-stationary linear and nonlinear systems as well as corresponding analytical solutions have been provided already in the 1960s by Kalman and Stratonovich, respectively, based on the generalization of some earlier results by Kolmogorov and Wiener. While analytical solutions do exist, their application to real world systems is often limited by high dimensionality of optimized models as well as the lack of empirical data for their proper parametrization. To resolve the above issues, in the past years numerous approaches have been suggested, such as different extensions of the Kalman filter based on the linearization of nonlinear effects in the system and so on.

Despite the above remark, I believe that this work is nevertheless publication worthy, given that for the great variety a practical scenarios that can be realized in real world complex systems, until now there is no universal solution that would overcome the above limitations of high dimensionality and relatively easy parametrization. Accordingly, any reasonable approach in this direction that is likely applicable to a wide range of real world complex systems and appears rather universal is of considerable interest. The Bayesian updating that is based on a solid theoretical background and can be relatively easily implemented clearly satisfies the above requirements.

The approach used here by the authors is largely based on splitting the overall model into low-level and high-level models describing the local and the global dynamics of the system, respectively. Generally, similar concepts are ubiquitous and can be found in literature since the early works on extended Kalman filtering where the "slow" and the fast "fluctuating" components are extracted

and processed separately leading to the overall design simplification. Among more recent approaches the so-called superstatistical concept where the rate of a simple process is modulated by a certain high-order variation model has to be mentioned, see, e.g. [Physica A 322, 267-275]. In recent years, this approach deserved a large number of applications to real-world complex systems including financial markets, sea levels, rainfall dynamics, internet traffic, biological polymer structures etc., see, e.g. [Phys Rev E 80 (3), 036108; Physica A 417, 18-28; Physica A 453, 173-183; J Phys A 49 (15), 154001; EPL 115 (1), 10008; Phys Rev E 94, 042305; Sci Rep 7, 43034, 46917] and references therein. Within this concept, a relatively simple low-level model, that often contains assumptions that are not valid at long scales, such as independence of extreme events etc., is superimposed by a high-level model that described the “slow” system state variations. While this approach is in general not new, it appears legitimate and in fact helps to make the model parameterization easier. Thus I suggest that the authors put their results in the context of the recent literature.

Moreover, while regular low-level model updating according to clear and objective criteria driven by Bayesian statistics in fact could lead to great performance at short scales, for truly complex systems it is essential that high-level models provide an accurate representation of their long-term dynamics. From the current version of the ms, it is not very clear how the authors are going to achieve this goal. While neglecting inherent long-range effects that are commonly evidenced by both linear and nonlinear long-range memory may be sometimes acceptable when analyzing quantities that are mainly dependent on the short-term dynamics, see, e.g. [Physica A 485, 48-60], in other cases (including long-term climate models) they lead to significant underestimations of the confidence intervals and thus to the excessive confidence in the model predictions that increase with increasing prediction time, see, e.g. [Nat Geo 7 (4), 246-247; Clim Dyn 46 (1-2), 263-271; PNAS 2017, E2998-3003]; other examples where neglecting long-range correlations may lead to spurious results include biological polymer structures [Sci Rep 7, 43034, 46917]. This issue should be clarified by the authors prior to publication.

Finally, as a general remark, for the validation of their approach, the authors go directly to the observational data from various complex systems, without validation using some simulated data. The disadvantage here could be that in the observational data the true dynamics can be extracted only from a posteriori evidence. That is not a problem for prediction of particular values, while this may be a problem for predicting distributions, quantiles etc., since their a posteriori estimates are also based on finite samples and, with the clear focus on short-term dynamics, often very small samples. At this point, the authors may like to comment on the accuracy of these estimates and how it is related to the model selection or potential prediction accuracy.

Particular questions

1. For the financial data, the authors use a simple autoregressive model for the short-term dynamics. It is common that on a long-term scale returns are considered as linearly uncorrelated, while exhibiting pronounced nonlinear memory, often represented by some models like the multiplicative random cascade or the multifractal random walk. Is the variability of the (extremely short-term) correlations really essential at minute resolution data that they need an AR1 component? Maybe it would be easier to reduce to random variations modulated by variable (and long-term correlated) volatility?

2. Related to previous question: the authors mention that the short-term correlations occur largely in response to previous significant perturbations, in particular abrupt price changes. In this case, the short-term correlated (decaying) perturbations could be just treated as a response to this abrupt change. In particular, it has been suggested very recently that the intraday price fluctuations seem to show a kind of a step response to the overnight jump [EPL 118, 18004]. Maybe similar principle can be used to describe response to any abrupt change, not just the overnight one.

3. As far as I understand it from the entire model design, it is capable of accurately predicting the system response to some extremes, but neither the extremes themselves nor some probabilistic characteristics of expected extremes? Is that correct?

4. For a more practical outlook, it would be interesting to see how the approach suggested by the authors would perform in the dynamic Value-at-Risk estimation widely used in applied economics such as finding best insurance strategies, that can be simply obtained as the corresponding estimated distribution quantile. While several approaches for the dynamic Value-at-Risk estimates have been suggested recently [Phys Rev E 80 (2), 026131; EPL 95 (6), 68002; Phys Rev E 94, 042305], all of them are based on various suboptimal updating strategies. While on the short-term the Bayesian updating theoretically should outperform other (suboptimal) updating strategies, it is always questionable whether the statistics would be sufficient for the accurate parametrization of the (more complex) quantities used in the Bayesian updating compared to alternative approach that utilizes simple quantities like intervals between consecutive events exceeding a certain magnitude, especially under high temporal resolution analysis. It is especially interesting how such estimates would compare on longer scales.

Summary

To summarize, the presented study is based on a solid theoretical background, clearly written and suitable for a general audience. It is commendable that the authors have implemented their solutions as a software package and made it available for the research community. The authors also provide several examples of the application of their approach to real-world complex systems.

Therefore this work is of general interest and deserves publication. However, there are several aspects of this work that leave open questions in the current form of the ms which should be clarified by the authors prior to publication. Moreover, the literature on the subject is only partially represented. The revised ms could put the research more in the context of alternative approaches to similar complex systems. Direct comparison could particularly emphasize the strength of the suggested approach as well as demonstrate its possible disadvantages more clearly.

Reviewer #1

My major concern is that the authors do not discuss the limitations of their methods. For example, the authors represent probability distributions by gridding parameter space. This effectively limits their application to models with only a few parameters. Consequently, they are forced to deal with simple, phenomenological models of complex processes. I do not think a more complex, mechanistic (or physics-based) model (that includes hundreds or thousands of possibly time-varying parameters) could be considered.

We fully agree and have added a paragraph and a supplementary figure (S1) explaining the limitations of our approach. Our grid-based approach is limited to models with up to 3 time-varying parameters, as a higher-dimensional grid takes up excessive memory during computation. One has to find a trade-off between the number of grid points (which influences the accuracy of parameter estimates and model evidence), and the computation time and memory usage. Supplementary Fig. 1 now gives an overview on the performance of the algorithm and illustrates the convergence of model evidence values and the ratio of model evidence values (the Bayes factor) for competing models.

Along these lines, I would like to see a frank discussion of what types of data and models can be reasonably considered within their framework.

We have now included a discussion of this important point in the main text. In the presented work, we deal with small datasets/short time series, with a number of time steps on the order of a few hundred. As the number of data points increases, the parameter distributions generally become narrower (which requires a finer spacing of grid points, which in turn needs more computational time). We have tested our method using a wide range of lattice constants and time series containing more than 10^4 data points. The computational time of our method scales linearly with the number of data points times the number of grid points. Hence, with a standard PC, the analysis of time series with up to $\sim 10^4$ data points and $\sim 10^4$ grid points for the time-varying parameters is unproblematic.

Regarding the class of supported models, our method relies on a factorizable likelihood function so that it can be computed as the product of the probability of each data point. The class of models which supports such a likelihood factorization includes not only models with independent observations (e.g. a Poisson process), but also auto-regressive models for which the current data point depends on past data points (e.g. a correlated random walk model). Not supported are models in which the current data point also depends on past parameter values, as in moving-average models. We address this restriction in more detail in the revised methods section.

Are there any underlying assumptions about the way parameters vary in time? For example, if the parameters were to vary with a characteristic time scale faster than data sampling frequency, would this pose any problems?

There are no underlying assumptions about the way parameters vary in time other than those included in the model. For example, the model may allow the parameters to jump abruptly from one time step to the next, but parameter-variations on a time scale faster than the sampling frequency cannot be resolved.

What if you use the wrong transformation between time steps?

Generally speaking, there are no “wrong” transformations between time steps, as long as the transformation is norm-conserving. Different transformations are, however, more likely or less likely to describe the data at hand. For example, the model evidence can be used to decide which transformation (e.g. assuming only gradual parameter changes, abrupt parameter changes, or both) fits a measured heterogeneous random walk trajectory best.

Related to the last point, I wondered why the authors chose a regular grid to represent the probability distribution? The ultimate goal is to approximate integrals, and I suspect performance would be improved by using Gauss points (i.e., quadrature).

Indeed, the performance of approximating integrals (in our case, the model evidence) would greatly benefit from using a non-regular grid and would open up the possibility of addressing models with more than 3 time-varying parameters. However, we also need to transform the probability distribution represented on the parameter grid in different ways. This transformation step is done repeatedly after evaluating each data point of the time series (in forward and backward direction), which is why the performance of those transformations is even more important to us. Currently implemented transformations include the

convolution with a Gaussian distribution and the Alpha-stable distribution. We rely on fast Python implementations of these convolutions, which in turn require a regular grid. In addition, we hope that other researchers will extend our software package *bayesloop* in the future, adding new transformations that specifically fit their modeling needs. Finally, we believe that the algorithm is easier to understand and to extend based on a regular grid.

Minor concerns:

In the opening sentences of the abstract, the authors write that seemingly chaotic dynamics are difficult to model because the model parameters vary in time. I think they mean to say that the model parameters "can vary in time." Although the difference is subtle, I would point out that the definition of "complex system" and "chaotic dynamics" is one of active, ongoing discussion. As it is currently written, it appears the authors are taking an ideological stand that this is their definition of "complex system" and "chaotic dynamics" when I believe they simply wish to introduce the specific problem they consider.

We fully agree and have corrected the text.

I appreciate that the authors' code is freely available on github. I would further recommend they get a dedicated DOI for the version of the code associated with this manuscript and cite it in their manuscript.

We agree and have obtained a DOI: [10.5281/zenodo.1193665](https://doi.org/10.5281/zenodo.1193665)

Occam's razor as a rule for model selection is mentioned several times. On this point, the authors may find work of Balasubramanian useful/interesting (Balasubramanian, Vijay. "Statistical inference, Occam's razor, and statistical mechanics on the space of probability distributions." *Neural computation* 9.2 (1997): 349-368). More recent (though unpublished) work of LaMont and Wiggins (arXiv:1706.01428) and Mattingly et al. (arXiv:1705.01166) attempt to take these ideas further.

Thank you for pointing out these papers. We now cite the work by Balasubramanian when we first mention Occam's razor, and the work of Mattingly et al. in the discussion of grid-based inference methods. This novel method could be utilized to build a grid-based inference method which not only provides uninformative priors but also the optimal parameter grid (or set of "atoms") for parameter inference, based on the available amount of data. The revised manuscript discusses these points as an outlook.

Reviewer #2

Overall this is interesting and relevant research, but the paper is not referring in a sufficient way to previous work in this general area. Basically, what the authors do is now well-established under the name "superstatistics", and many groups are working on similar topics, but these other groups are not cited at all. So I think it is essential to mention in the introduction and abstract that this paper is basically considering ***superstatistical*** complex systems, not general complex systems which can be anything. The authors are looking at systems where certain parameters that are usually kept constant fluctuate on a long time scale, and they have a simple local model that depends on these parameters, and they provide useful methods for local model validation. But one misses the most essential papers on superstatistical methods in the list of references, such as Beck and Cohen *Physica A* 322, 267 (2003), or Beck, Cohen, Swinney (*PRE* 72, 056133 (2005)) which extracts superstatistical parameter distributions (similar as the ones considered in this paper) from a given time series. Surprisingly, in their previous work (ref [10]) the authors mention superstatistics in the title of their paper, but they completely ignore it in the current paper, for reasons unclear to me.

We agree that the connection to superstatistics should be made much more clear. The revised manuscript now puts our work into a superstatistical context, starting with the first sentence of the abstract.

The authors should add a paragraph on superstatistics in their introduction and put their work into context. It is essential to cite more recent systematic papers on superstatistical dynamics, such as the important work by Chechkin et al in *Phys. Rev. X* 7, 021002 (2017) or the new work by Schaefer et al in *Nature Energy* which appeared in January 2018 and which applies it to yet another example, frequency fluctuations in power grids. Otherwise, the current manuscript completely misses out on new recent important developments in this field.

In the revised introduction, we added a paragraph describing the superstatistical approach to the analysis of complex systems. There, we also cite relevant literature (including Chechkin et al., Beck et al., Beck and Cohan et al., and also Schaefer et al. at a later point) for readers to follow-up in more detail.

Finally, the authors should better explain what is really new in their approach, as compared to previous

approaches. I do appreciate there is something new: The systematic Bayesian approach, and also the fact that they typically study 2 superstatistical parameters rather than 1, such as volatility and correlation, whereas previous papers mainly studied just the volatility (though papers such as Jizba et al., Physica A 493, 29 (2018) also study several superstatistical parameters in financial time series).

The revised manuscript (Introduction) now explicitly states the two novel extensions to the superstatistical approach: First, we reconstruct the complete temporal evolution of the time-varying model parameters, to not only determine how frequently certain parameter values are realized, but to further pin-point when parameter values change. Second, and most important, our method objectively selects between different time-varying parameter models.

Concerning the four specific applications in this paper, people have worked on similar applications before, but again references to previous work are missing. The Poisson process model with varying rate as displayed in Fig. 2b has already been studied previously as a simple model for (not mining accidents but) train delay statistics, see Briggs et al Physica A 378, 498 (2007).

Thank you for pointing out the work by Briggs et al. The revised manuscript discusses this and other papers on rate fluctuations in Poisson processes and puts our work into the proper context. In particular, our example focuses not on rate fluctuations alone, but how a conventional change-point model may benefit from additional Gaussian rate fluctuations.

Cancer cell migration has been previously discussed in a superstatistical context by Chen et al in Physica A 387, 3126 (2008), so when discussing the heterogeneity in Fig.3 this previous work could be cited (though the approach there is a bit different)...

Thank you for pointing out the work by Chen et al. We have added a sentence to the relevant paragraph, referring to works on heterogeneity in cancer progression on different spatio-temporal scales.

...and then it should be explained what is new as compared to ref [10], the previous work of the authors.

In the previous work, the magnitude of the parameter variations was set to a fixed value which was estimated from simulated data. Here, the parameter dynamics are optimized based on the data, using the model evidence. This crucial improvement is noted in the introduction.

Stock market fluctuations have been studied by many authors in a superstatistical context before, so the authors should make clearer what is new in their approach, as compared to references cited in Jizba et al Physica A 2018.

Thank you for pointing out the work by Jizba et al. We added a paragraph to the finance example to relate our two-parameter model to existing stochastic volatility models and relevant superstatistical analyses noted in Jizba et al.

The abrupt parameter changes in Fig. 5d, which occur at certain times of the day for share price index movements, are also seen in electricity markets, see e.g. Fig.1 of Schaefer et al in Nature Energy 2018, where they are caused by trading intervals.

Thank you for pointing out the work by Schaefer et al. The revised manuscript refers to this additional example of temporal clustering of trading-induced anomalies.

Global warming has also been studied in a superstatistical context before, see. e.g. Yalcin et al Physica A 392, 5431 (2013), but the approach in the current manuscript in Fig. 6 is different---the authors should better explain that they look at critical points where a drastic change happens, which is new and was not done in previous work.

We agree, the revised manuscript now mentions the previous superstatistical work on global warming and further emphasizes the identification of critical points.

A minor comment: What is atan^2 in eq.(18)?

atan^2 represents an inverse tangent that is confined to the interval $(-\pi, \pi]$. It is widely employed in computer languages and provides a convenient way of computing e.g. turning angles from subsequent velocity vectors, while the "standard" arctangent function cannot distinguish between diametrically opposite directions. For clarification, the revised manuscript includes a proper definition based on the arctangent function.

Reviewer #3

The approach used here by the authors is largely based on splitting the overall model into low-level and high-level models describing the local and the global dynamics of the system, respectively. Generally, similar concepts are ubiquitous and can be found in literature since the early works on extended Kalman filtering where the “slow” and the fast “fluctuating” components are extracted and processed separately leading to the overall design simplification. Among more recent approaches the so-called superstatistical concept where the rate of a simple process is modulated by a certain high-order variation model has to be mentioned, see, e.g. [Physica A 322, 267-275]. In recent years, this approach deserved a large number of applications to real-world complex systems including financial markets, sea levels, rainfall dynamics, internet traffic, biological polymer structures etc., see, e.g. [Phys Rev E 80 (3), 036108; Physica A 417, 18-28; Physica A 453, 173-183; J Phys A 49(15), 154001; EPL 115 (1), 10008; Phys Rev E 94, 042305; Sci Rep 7, 43034, 46917] and references therein. Within this concept, a relatively simple low-level model, that often contains assumptions that are not valid at long scales, such as independence of extreme events etc., is superimposed by a high-level model that described the “slow” system state variations. While this approach is in general not new, it appears legitimate and in fact helps to make the model parameterization easier. Thus I suggest that the authors put their results in the context of the recent literature.

Thank you for raising this important issue. The revised manuscript now includes an introductory paragraph about superstatistics to put our work into proper context and refers to relevant literature for all presented applications of our method. Thank you also for the suggested literature that we now cite in the revised manuscript.

Moreover, while regular low-level model updating according to clear and objective criteria driven by Bayesian statistics in fact could lead to great performance at short scales, for truly complex systems it is essential that high-level models provide an accurate representation of their long-term dynamics. From the current version of the ms, it is not very clear how the authors are going to achieve this goal. While neglecting inherent long-range effects that are commonly evidenced by both linear and nonlinear long-range memory may be sometimes acceptable when analyzing quantities that are mainly dependent on the short-term dynamics, see, e.g. [Physica A 485, 48-60], in other cases (including long-term climate models) they lead to significant underestimations of the confidence intervals and thus to the excessive confidence in the model predictions that increase with increasing prediction time, see, e.g. [Nat Geo 7 (4), 246-247; Clim Dyn 46 (1-2), 263-271; PNAS 2017, E2998-3003]; other examples where neglecting long-range correlations may lead to spurious results include biological polymer structures [Sci Rep 7, 43034, 46917]. This issue should be clarified by the authors prior to publication.

Thank you for pointing this out to us. Indeed, long-range correlations represent a limitation for our method. Long-range correlations (with a power-law correlation function) cannot be modeled by our iterative inference approach, as the likelihood function only factorizes for models with independent observations or short-range correlated observations (like in auto-regressive processes). The revised manuscript discusses this limitation in the Methods section, Discussion, and Results (in the context of the climate example). We further cite the relevant literature.

Finally, as a general remark, for the validation of their approach, the authors go directly to the observational data from various complex systems, without validation using some simulated data. The disadvantage here could be that in the observational data the true dynamics can be extracted only from a posteriori evidence. That is not a problem for prediction of particular values, while this may be a problem for predicting distributions, quantiles etc., since their a posteriori estimates are also based on finite samples and, with the clear focus on short-term dynamics, often very small samples. At this point, the authors may like to comment on the accuracy of these estimates and how it is related to the model selection or potential prediction accuracy.

Thank you for pointing out this important issue. We added 2 supplementary figures (S4 & S5) that provide additional verification of inferred distributions and correlation coefficients of time-varying parameters. In particular, we applied bootstrapping to simulate “new” datasets for the coal-mining data (110 data points) and the cancer cell trajectories (with a minimal length of 25 data points), based on the inferred parameter estimates of the actually observed data. Repeating the analysis with these simulated datasets, we obtain error estimates for all inferred quantities.

Particular questions

1. For the financial data, the authors use a simple autoregressive model for the short-term dynamics. It is common that on a long-term scale returns are considered as linearly uncorrelated, while exhibiting pronounced nonlinear memory, often represented by some models like the multiplicative random cascade or the multifractal random walk. Is the variability of the (extremely short-term) correlations really essential at

minute resolution data that they need an AR1 component? Maybe it would be easier to reduce to random variations modulated by variable (and long-term correlated) volatility?

Indeed, when we repeat the analysis of the financial data assuming no short-term correlation, we find that the heavy-tailed distribution of returns of SPY can be explained equally well, based only on a time-varying volatility parameter. We added Supplementary Fig. 7 for a direct comparison of the one- and two-parameter model. However, we still believe the two-parameter model including short-term correlations offers additional insights. In particular, we observe a small but significant cross-correlation between the short-term correlations and volatility with a lag-time of 45min (i.e., changes in the correlation of subsequent returns tend to precede changes in volatility by 45 min). Note that this long-range correlation cannot be explicitly included in our model structure, but instead is identified from the posterior mean values of the two parameters.

2. Related to previous question: the authors mention that the short-term correlations occur largely in response to previous significant perturbations, in particular abrupt price changes. In this case, the short-term correlated (decaying) perturbations could be just treated as a response to this abrupt change. In particular, it has been suggested very recently that the intraday price fluctuations seem to show a kind of a step response to the overnight jump [EPL 118, 18004]. Maybe similar principle can be used to describe response to any abrupt change, not just the overnight one.

That is a very interesting point. We added Supplementary Fig. 8, which shows the average response functions of the short-term correlation and volatility parameter after an “anomalous” abrupt price change. Here, we used our real-time model selection analysis example to identify “anomalous” price changes with a risk of 5% of rendering previously learned parameter estimates useless. After such an event, volatility is on average increased by 60% and subsequently falls off according to a power-law. The short-term correlation, however, shows no consistent upward or downward variation. Therefore, the short-term correlation cannot be treated in a systematic way as a response to an abrupt change.

3. As far as I understand it from the entire model design, it is capable of accurately predicting the system response to some extremes, but neither the extremes themselves nor some probabilistic characteristics of expected extremes? Is that correct?

That is correct. Our method can only estimate the system’s parameter values and how they change over time (i.e. how they are distributed), but our method cannot predict or extrapolate the distribution of the observed data other than through simulation. To illustrate this point, we simulate price fluctuations (Fig. 4f) based on the superstatistical distribution of (the previously inferred) short-term correlation and volatility (Fig. 4e). When we compare the simulated price fluctuations with the actually measured price fluctuations, we find good agreement in their distribution, including for rare events.

4. For a more practical outlook, it would be interesting to see how the approach suggested by the authors would perform in the dynamic Value-at-Risk estimation widely used in applied economics such as finding best insurance strategies, that can be simply obtained as the corresponding estimated distribution quantile. While several approaches for the dynamic Value-at-Risk estimates have been suggested recently [Phys Rev E 80 (2), 026131; EPL 95 (6), 68002; Phys Rev E 94, 042305], all of them are based on various suboptimal updating strategies. While on the short-term the Bayesian updating theoretically should outperform other (suboptimal) updating strategies, it is always questionable whether the statistics would be sufficient for the accurate parametrization of the (more complex) quantities used in the Bayesian updating compared to alternative approach that utilizes simple quantities like intervals between consecutive events exceeding a certain magnitude, especially under high temporal resolution analysis. It is especially interesting how such estimates would compare on longer scales.

Thank you for pointing us to the VaR as a commonly employed risk estimate. We added two supplementary figures (S9 & S10) which explore the idea of a Bayesian VaR estimation. Furthermore, we provide a direct comparison of the Bayesian VaR estimates with the moving-window approach and the return-interval-approach, based on day-to-day returns of the fund SPY. In summary, we find Bayesian VaR estimation to be a viable alternative to the return-interval-approach.

REVIEWERS' COMMENTS:

Reviewer #1 (Remarks to the Author):

I thank the authors for taking the time to respond carefully to my comments. I am satisfied they have answered my questions and concerns, and I recommend the manuscript for acceptance.

Reviewer #2 (Remarks to the Author):

The revised version has significantly improved. The paper is now much clearer and it is easier to understand what the

authors are doing as compared to previous work, and why it is

relevant. All comments raised in my previous report were successfully taken into account. I recommend acceptance.

Reviewer #3 (Remarks to the Author):

As I already wrote in the first review round, this ms is of general interest and deserves publication. The authors propose an interesting new modification of the well established Bayesian updating approach and highlight its important applications to various complex systems. I am very pleased that the authors took considerable effort to answer all the queries that arose in the first review round and added several very transparent examples that further highlight both the potential and the limitations of the proposed methodology.

In the view of the latter, given that the authors state clearly (on page 15 of the revised ms) that their model does not support several classes of dynamical models including those with long-range correlations, I believe this important limitation concerning the classes of supported systems could be briefly noted already in the abstract. While admitting that under certain conditions the proposed approach performs well also for systems exhibiting long-range correlations, especially under pronounced domination of short-term influences, as nowadays many groups worldwide advocate

the importance of long-range correlations, including those classes of complex dynamical systems considered by the authors such as finance and climate (also considering several papers now quoted in the revised ms), I believe such a remark is essential.

Besides that minor remark, I am pleased to recommend the revised ms for publication in its present form.

Reviewer #3

As I already wrote in the first review round, this ms is of general interest and deserves publication. The authors propose an interesting new modification of the well established Bayesian updating approach and highlight its important applications to various complex systems. I am very pleased that the authors took considerable effort to answer all the queries that arose in the first review round and added several very transparent examples that further highlight both the potential and the limitations of the proposed methodology.

In the view of the latter, given that the authors state clearly (on page 15 of the revised ms) that their model does not support several classes of dynamical models including those with long-range correlations, I believe this important limitation concerning the classes of supported systems could be briefly noted already in the abstract. While admitting that under certain conditions the proposed approach performs well also for systems exhibiting long-range correlations, especially under pronounced domination of short-term influences, as nowadays many groups worldwide advocate the importance of long-range correlations, including those classes of complex dynamical systems considered by the authors such as finance and climate (also considering several papers now quoted in the revised ms), I believe such a remark is essential.

Besides that minor remark, I am pleased to recommend the revised ms for publication in its present form.

The revised abstract now explicitly states that our method applies to short-range correlated time series.